# Modelling dependence and coincidence of storm surges and high-tide: Methodology, discussion and recommendations based on a simplified case study in Le Havre (France)

Amine Ben Daoued [1], Yasser Hamdi [2], Nassima Mouhous-Voyneau [1], Philippe Sergent [3]

[1] Sorbonne University, Université de Technologie de Compiègne, 60203 Compiègne, France
[2] Institute for Radiological Protection and Nuclear Safety, 92 262 Fontenay-Aux-Roses, France
[3] Centre d'étude et d'expertise sur les risques, l'environnement, la mobilité et l'aménagement, France

*Correspondence to:* Y. Hamdi (yasser.hamdi@irsn.fr)

**Abstract.** Coastal facilities such as nuclear power plants (NPPs) have to be designed to withstand extreme weather conditions and must, in particular, be protected against coastal floods because it is the most important source of coastal lowlands inundations. Indeed, considering the combination of tide and extreme storm surges (SSs) is a key issue in the evaluation of the risk associated to coastal flooding hazard. Most existing studies are generally based on the assumption that high tides and extreme SSs are independent. While there are several approaches to analyze and characterize coastal flooding hazard with either extreme SSs or sea levels, only few studies propose and compare several approaches combining the tide density with the SS variable. Thus this study aims to develop a method for modelling dependence and coincidence of SSs and high tide. In this work, we have used existing methods for tide and SS combination and tried to improve the results by proposing a new alternative approach while showing the limitations and advantages of each method. Indeed, in order to estimate extreme sea levels, the classic joint probability method (JPM) is used by making use of a convolution between tide and the skew storm surge (SSS). Another statistical indirect analysis using the maximum instantaneous storm surge (MSS) is proposed in this paper as an alternative to the first method with the SSS variable. A direct frequency analysis using the extreme total sea level is also used as a reference method. The question we are trying to answer in this paper is then the coincidence and dependency essential for a combined tide and SS hazard analysis. The results brought to light a bias in the MSS based procedure comparing to the direct statistics on SLs and this bias is more important for high return periods. It was also concluded that an appropriate coincidence probability concept, considering the dependence structure between SSs, is needed for a better assessment of the risk using the MSS. The city of Le Havre in France was used as a case study. Overall, the example has shown that the return levels (RLs) estimates using the MSS variable are quite different from those obtained with of the method using the SSSs, with acceptable uncertainty. Furthermore, the shape parameter is negative form all the methods with a much heavier tail when the SSS and the extreme sea levels (ESLs) are used as variables of interest.

**Key-words:** Coastal flooding, Combination, Joint Probability Method, Convolution, Dependence, Coincidence

## 1. Introduction

Like any other urban facilities, Nuclear Power Plants (NPPs) can be subject to external influences and aggressions such as extreme environmental events (river and/or marine flooding, heat spells, etc.). Both nuclear and urban facilities have to be designed to withstand extreme weather conditions. During the last few decades, France has

experienced several violent storms (the great storm of 1987, Lothar and Martin cyclone in 1999, Klaus in 2009 and
Xynthia in 2010, for instance) that gave rise to exceptional SSs. Many coastal facilities was partially or completely
flooded when storm Martin struck the French coast in 1999. A combination of an exceptional SS, of a high tide and
high waves induced by strong winds led to the overflow of many dikes which were not designed for such a
concomitance of events. In the nuclear safety field for instance, a guide to protection, including some fundamental
changes in the assessment of flood risks, has therefore been produced by the Nuclear Safety Authority (ASN, 2013).
However, to be conservative, approaches used in the guide are deterministic which do not take into account all the
local specificities of each site. The safety demonstration and protections are periodically reviewed to ensure
compliance with the increased safety requirements. The present work could be used to enrich safety verification
approaches, by proposing other approaches and confronting them to the reference method currently used in the
guide. To supplement knowledge which can be acquired from the deterministic method, the probabilistic approach
has been identified as an effective tool for assessing risk associated with hazards as well as for estimating
uncertainties.
The first probabilistic study in the nuclear safety field was conducted in the United States in 1975 (US-NRC, 1975).
This report focused on estimating the probability of occurrence of meltdown accidents with associated radiological
consequences. Currently, probabilistic approaches are applied in several fields such as medicine, chemical industry,
insurance and aeronautics. Many studies have already been conducted for the seismic hazard (IAEA, 1993; Beauval,
2003; Gupta, 2007), the tsunami hazard (IRSN, 2015), and other climatic hazards such as tornadoes (US-NRC,
2007). There are not many probabilistic studies yet in the fields of climate and hydrometeorology, as it is an approach
barely used. In fact, very few researches and developments are explicitly referred by their authors as conclusive and
operational. Probabilistic Flood Hazard Assessment (PFHA) is identified by Bensi and Kanney (2015) as a first step
in a Probabilistic Risk Assessment (PRA). According to the authors, it is an evaluation of the probabilities that one
or more parameters representing the severity of the external flood (water level, duration, and associated effects) are
exceeded in a site of interest. Also, the authors discuss the joint probability method (JPM) as an alternative to existing
deterministic and statistical methods such as the Empirical Simulation Technique (EST). Kügel (2013) proposed a
methodology for characterizing the external flood hazard for nuclear sites located alongside rivers and the
articulation of this Hazard study with a flooding Probabilistic Safety Assessment (PSA).
It is a common belief today that the probability of failure, over an infrastructure lifetime is one of the most important
pieces of information an engineer can communicate. The estimation of the probability of exceeding an extreme event
should be based on the combination of all flood sources (e.g. Pluvial, fluvial and coastal floods) which are most
often dependent because they are induced by the same storm. Mostly, a flood phenomenon can be characterized by
several explanatory variables, some of which are correlated. The problem of the surge-tide interactions has been
addressed in the literature for many regions and with different approaches (Coles and Tawn, 2005; Gouldby et al.,
2014; Pirazzoli, 2007; Idier et al., 2012; Idier et al., 2019). It was shown that tide–surge interactions can be relevant
in several regions. The tide–surge interactions at the Bay of Bengal (corresponding to the effect of the tide on
atmospheric surge and vice versa) were analyzed by Johns et al., (1985) and Krien et al., (2017). They showed that
tide–surge interactions in shallow areas of this large deltaic zone are in the range ±0.6m occurred at a maximum of
1 to 2 hours after low tide. Similar results were obtained by Johns et al. (1985), Antony and Unnikrishnan (2013)
and more recently Hussain and Tajima (2017). Focusing on the English channel, Idier et al. (2012) used shallow
water model to make surge computations with and without tide for two selected events (November 2007 North Sea

and March 2008 Atlantic storms). The authors concluded that the instantaneous tide–surge interaction are significant in the eastern half of the English Channel, reaching values of 74 cm in the Dover Strait, which is about half of maximal storm surges induced by the same events. They also concluded that Skew surges are tide-dependent, with negligible values (less than 5 cm) over a large portion of the English Channel, but reaching several tens of centimeters in some locations such as the Isle of Wight and Dover Strait. More recently, Idier et al. (2019) have investigated the interactions between the sea level components (sea level rise, tides, storm surges, etc.) and the tide effect on atmospheric storm surges is among the main interactions investigated in their review. The authors stated that the studies, and other ones, converge to highlight that tide–surge interactions can produce tens of centimeters of water level at the coast.

On the other hand, there are some phenomena which are described by other explanatory phenomena. The case of multi-components phenomena, that will receive our attention in the present paper, is the coastal flooding which is a combination of tides with SSs. Indeed, the SS is one of the main drivers of coastal floods. It is an abnormal rise of water generated by a storm (low atmospheric pressure and strong winds), over and above the predicted tide. It should be noted that the effect of waves (runup and setup) on total water level is not discussed in the present paper. Extreme storms can produce high sea levels, especially when they coincide with high tide. The skew storm surge SSS is a sea level component which is often considered as the fundamental input or the quantity of interest for statistical investigations of coastal hazards. It is the difference between the highest observed level and the highest predicted one, for a same high tide. These maximum levels can occur at slightly different times.

As more than one explanatory variable are often used in a PFHA and in case these variables are dependent, the dependency structure must be modeled and a consistent theoretical framework must be introduced for the calculation of the return periods and design quantiles with multivariate analysis based on Copulas (e.g. Salvadori et al., 2011). Indeed, numerous studies have shown that, in case of multivariate hazards, a univariate frequency analysis does not allow to estimate in a complete way the probability of occurrence of an extreme event (Chebana and Ouarda, 2011; Hamdi et al., 2016). According to Salvadori and De Michele (2004), modelling the dependency allows a better understanding of the hazard and avoids under/over-estimating the risk. Unsurprisingly, some ideas have been proposed in the literature for combining tides and SSs and to help address such an important issue. JPM is an indirect method that made an improvement in addressing the main limitations of the direct methods (e.g. the annual maxima method (AMM) and the r-largest method (RLM)) (Haigh et al., 2010). Several studies refer to the JPM for the probabilistic characterization of storms (Batstone et al., 2013; Haigh et al., 2010; Pugh and Vassie, 1978; USACE, 2015). Tawn and Vassie (1989) proposed a Revised JPM (RJPM) in which the distribution of surges is composed by a left tail defined by an empirical method and a right tail defined by frequency analysis. Dixon and Tawn (1994) made some modifications on the Revised JPM and proposed a new model to take into account the interaction between instantaneous SS and tide. Recently, Haigh et al. (2010) showed the advantages of indirect methods (i.e. JPM, Revised JPM) compared to direct ones (i.e. AMM and RLM). More recently, Kergadallan et al. (2014) proposed an extension of the model proposed by Dixon and Tawn (1994) using skew storm surges (SSSs) at 19 French harbours along the Atlantic and English Channel coasts of France. The authors have used two different approaches (the seasonal dependence and the interaction between SSs and tides) to study the dependence of the SSs on the tides with three methods (the seasonal approach, Dixon and Tawn (1994) model and the revisited Dixon and Tawn model). It was concluded that the interaction between SSSs and high tides affect more significantly the results than the seasonal dependence for more than one-half of the harbours.

Some other studies have been proposed in the literature to tackle the PFHA. The most important contribution
proposes two methods. The first estimates extreme sea levels (ESLs) with the JPM (Pugh and Vassie, 1980). Indeed,
this approach combines separated frequency distributions for the tide (usually deterministic and exact) and the SS
(frequency analysis based on the extreme value theory). It is a calculation of the convolution based on the tidal levels
density function and of a distribution function of SSs. Duluc et al., (2012) have shown that the quality of the results
from this convolution approach for small return periods is questionable. The second procedure uses the data of
observed maximum water levels (Chen et al., 2014; Haigh et al., 2014; Huang et al., 2008). This approach was
recommended by FEMA's guideline (FEMA, 2004) for coastal flood mapping, in which the GEV model is
recommended to conduct the frequency analysis of extreme water levels, if long-term datasets are available. Based
on the regional observations, the process of estimation of extreme water levels uses an adequate frequency analysis
model to estimate the distribution parameters, the desired return levels (RLs) and associated confidence intervals.
Overall, our goal is to build on the approaches and developments proposed in the literature and revive the debate as
to how researchers and engineers can combine tide with SS to estimate extreme sea levels. This goal is in line with
the recent literature (e.g. Idier et al., 2012, Kergadallan et al., 2014) challenging the use of the SSS and clearly
demonstrates the importance of using the maximum instantaneous surge (MSS) instead. In order to achieve this goal,
a third fitting procedure to estimate extreme sea levels using the MSS between two consecutive high tides is
introduced with an application so that it can be compared with the two first procedures. Mazas et al., (2014) proposed
a review of tide-surge interaction methods and applied a POT frequency model (with the GPD and Poisson
distribution functions) to the family of JPM-type approaches for determining extreme sea level values in a single
case study (Brest). The authors focused on the use of a mixture model for the surge component, which allows
probabilities to be quantified for the entire range of sea level values, not just for the extreme ones, which is not the
case here in the present paper.
The paper is organized as follows. The section 2 takes up the two fitting procedures proposed in the literature (the
JPM with a convolution between tides and SSSs and the frequency analysis directly on sea levels) and proposes a
new one based on the convolution between tides and MSSs. In section 3, the fitting procedures are applied on the
observed and predicted sea levels at the Le Havre tide gauge in France used as a case study. One of the most
important features of this case study is the fact that the lower parts of Le Havre city are likely to be flooded by
coastal floods and that the region has experienced important storms during the last few decades.

## 2. Methods

Tide and SSs are usually the subject of a statistical study to determine the probability of exceeding the water level
cumulating the two phenomena. Indeed, the SS is the main driver of coastal flood events. It is an abnormal rise of
water generated by a storm, over and above the predicted tide. As it would be analyzed later in the discussion section,
the dependency, in an extreme value context, is analyzed but not considered to combine the phenomena in the present
work. Indeed, as mentioned in the introductory section and as it will be discussed later in this paper, extreme levels
such as MSSs and high tides may be only very weakly dependent.
On the other hand, it is commonly known that the tidal signals can be predicted, and are not aleatory like the SSs.
What is somewhat odd in the present work is that one thus seeks to combine a distribution function of random
variable with a density of tide which is deterministic. In order to estimate extreme sea levels, a JPM is used by
making use of a convolution between tide and SSs. So the question that arises here is which variable of interest can
be used to better characterize coastal flooding? Three variables are then proposed: (i) the SSS; (ii) the MSS and (iii)
the extreme sea level. The theoretical basis for the fitting procedures using these variables is addressed in the
following subsections.
Relative to some chosen datum, each hourly observed sea level $Z(t)$, may be considered as the sum of its tide $X(t)$
and storm surge component $Y(t)$, i.e.:
$$Z(t) = X(t) + Y(t) \tag{1}$$
Thus if the probability density functions of the tidal and surge components are $f_X(x)$ and $f_Y(y)$ respectively then
the probability density function $f(z)$ of $z$, under the assumption that the tide and surge components are independent,
is:
$$f_Z(z) = \int_{-\infty}^{+\infty} f_X(x) \times f_Y(z - x)\, dx \tag{2}$$
As it can be seen in equation 2, the dependence on time, $t$, is omitted when replacing $X(t)$ by $X$, $Y(t)$ by $Y$, and
$Z(t)$ by $Z$. This implies a stationarity assumption for the involved time series. The hourly SS is often considered as
a stationary stochastic process, since meteorological and seasonal effects give rise to series of SSs randomly
distributed in time, but this is not the case of the hourly theoretical tide signals. It should also be noted that for the
case Le Havre the residual part as the surges is not the only one and despite the fact that it is the dominant component,
the stochastic signal also contains the fluvial effects.
**2.1 Joint SSS - tide probabilistic method**
This method is based on the decomposition of the sea level into a sum of two contributions: the tide which is
evaluated theoretically and the aleatory component SS obtained by subtracting the predicted tide from the observed
sea level. Extreme storms can produce high sea levels, especially when it occurs simultaneously with high tide. The
SSS is a sea level component which is often considered as the fundamental input for statistical investigations of
coastal hazards. It is defined as the difference between two observed and predicted maximums and is not impacted
by the shift of the two signals which may be biased (see figure 1). As shown in the left panel of figure 2, the SSS is
defined herein as the difference between the highest observed level and the highest predicted one, for a same high
tide (see equations 1 and 2). Further noteworthy features of SSSs are its occurrence with a high tide. Indeed, a SSS
occurring with a high tide is likely to induce a high sea level. Thus, for safety requirements, SSS is the most often
used in the literature Kergadallan et al. (2014).
Still, even if this procedure uses the suitable variable of interest, it has its limitations. Indeed, it is not uncommon
that the MSS, which can occur randomly somewhere between two consecutive tides, is greater than the SSS.
Widening the window around the high tide, in which extreme SSs are extracted, could improve frequency estimation
of extreme sea levels. When this window is maximum (12 hours, for instance), the variable of interest naturally
becomes the MSS. Moreover, it was demonstrated in the literature that the tide and SSS interaction at high tide
cannot be neglected (Kergadallan et al., 2014).

## 2.2 Joint MSS - tide probabilistic method

The right panel of figure 2 illustrates the case of an instantaneous SS signal, the variables would be the MSS and the high tide $M_n$. As mentioned in the previous section, the MSS can occur randomly somewhere in a tide cycle. One of the most important features of MSS is that it is more informative than the SSS. Indeed, the MSS covers the whole instantaneous SS signal. This feature makes the MSS a variable particularly useful for carrying out a PFHA exploring the entire tidal signal, not only the high tide.

## 2.3 Inference with the ESL: the reference method

For comparison purposes, we also analyzed sea levels signals for which we focused our attention on the frequency analysis on extreme sea levels without decomposing them into tides and surges. This yields to direct statistics and estimates of the RLs without combining tides and surges. The intent of this analysis is only to illustrate and obtain results that can serve as a reference for the comparison of the joint probability procedures. The maximum sea level between 2 high tide values is the variable of interest used for this reference procedure.

## 2.4 The sampling method

The Peaks-Over-Threshold (POT) sampling method is used conduct the frequency analyses in the present work. Commonly considered as an alternative to the annual maxima method, the POT method models the peaks exceeding a relatively high threshold. The distribution of these peaks converge to the Generalized Pareto Distribution (GPD) theoretical distribution. In addition, the threshold leads to a sample more representative of extreme events (Coles, 2001). However, the threshold selection is subjective and an optimal threshold is difficult to obtain. Indeed, a too low threshold can introduce a bias in the estimation because some observations may not be extreme data and this violates the principle of the extreme value theory. On another hand, the use of a too high threshold reduces the sample size (Hamdi et al., 2014).

On the other hand, all the simulations were carried out within the R environment (open source software for statistical computing: http://www.r-project.org/). The SeaLev library, developed by the French Institute for Radiological Protection and Nuclear Safety (IRSN), was used for the standard approach involving the convolution of the probability density functions of the tidal and surge heights to obtain the distribution of total sea levels. The frequency analyses were performed with the Renext library also developed by IRSN (IRSN and Alpstat, 2013). The Renext package was specifically developed for flood frequency analyses using the Peaks-Over-Threshold (POT) method.

## 3 Case study and data

The city of Le Havre is an urban city in the Seine-Maritime department, on the English Channel coast in Normandy (France). It is a major French city located in northwestern France. A map showing the location of the Le Havre city in France can be found in figure 3. The name Le Havre means "the harbour" or "the port". The port of Le Havre is, moreover, among the largest in France. For these reasons, the city of Le Havre remains deeply influenced by its maritime traditions.

Due to its location on the coast of the Channel, the climate of Le Havre is temperate oceanic. Days without wind are
rare. There are maritime influences throughout the year. According to the meteorological records, precipitation is
distributed throughout the year, with a maximum in autumn and winter. The months of June and July are marked by
some relatively extreme storms on average 2 days per month. One of the characteristics of the region is the high
variability of the temperature, even during the day. The prevailing winds are from north-northeast for breezes and,
from the southwest sector for strong winds.
The joint tide-surge probability and the frequency analysis of extreme sea levels are performed on the city of Le
Havre. The 1971-2015 observed and predicted hourly sea levels recorded at the port of Le Havre were provided by
the French Oceanographic Service (SHOM - Service Hydrographique et Océanographique de la Marine). Figure 4
shows the sea level time series of Le Havre, as well as the studied extreme SSs (SSSs and MSSs). One of the most
important features of Le Havre is the fact that it is subject to marine submersions and instabilities of coastal cliffs
(Elineau  et al., 2013; Elineau et al., 2010; Maspataud et al., 2016). In particular, the lower part of the city (Saint-
François district, for instance) is likely to be flooded by marine and pluvial floods. Data characteristics are shown
in the table 1. These data were first processed to keep only common periods containing a minimum of gaps. The
choice of the variables to be probabilized is done at this stage.
**4. Results**
Since we need to get comparable annual rates of extreme sea level events, the POT threshold selection process has
been adapted to meet this criterion and the thresholds are, even though, checked regarding the stability graphs of the
GPD parameters estimated with the maximum likelihood method The POT model characteristics (threshold and
associated average number of events per year) are presented in Table 2. The stability graphs for threshold selection
are presented in Figure 5.
The main results of the joint surge-tide probability method, with the SSS and MSS based fitting procedures, and the
results of the direct frequency analysis of the extreme sea levels as well, with all the diagnostics are presented in
terms of RL plots, estimates of the quantiles of interest and associated 95% confidence intervals. In these results,
the main focus was set to the 10-, 50-, 100- and 1000-year sea level RLs. Prior to the application of the JPM, the
SSSs and MSSs are calculated first from observed and predicted sea levels. The results of the application on the Le
Havre are summarized in table 3 and presented in figure 6.
The RL estimates obtained with the MSS based convolution are quite different from those of the one based on SSSs.
The results of the calculation of confidence intervals (with the delta method) are presented with transparent polygons
in figure 6 and in table 3 as well. As it can be noticed, the confidence intervals are relatively narrow. Indeed, the
relative width of the intervals around the 1000-year RL obtained with reference method, did not exceed 12%. Better
yet, the confidence intervals are narrower when using the joint probability procedures. It is interesting to note that
the delta method (Ver Hoef, 2012) is a classic technique in statistics for computing confidence intervals for functions
of maximum-likelihood estimates. The variance of RL estimates are calculated using an asymptotic approximation
to the normal distribution. Furthermore, it can be seen in figure 6 that for a given RL, the return period given by the
MSSs-based procedure is much lower than that given by the one based on the SSSs. The RLs are thus more
frequently (i.e. on average 10 times more frequently) exceeded randomly in a tidal cycle (i.e. as the MSS can occur
randomly somewhere inside a tidal cycle) than at the high tide moment (i.e. if we suppose that SSS often occurs at
the high tide moment).
It is noteworthy that the shape parameter $\xi$ of the General Pareto Distribution (GPD) is negative for all the cases
(i.e. $\xi = -0.2$; $\xi = -0.07$ and $\xi = -0.12$ for the SSS, MSS and ESL based fitting procedures, respectively).
This parameter governs the tail behavior of the GPD. The right tail of the distribution is much heavier for the
procedures using SSSs and the ESLs than for the one using MSSs.

**5 Discussion**

To objectively evaluate the merits and shortcomings of each of the methods described in section 2, the associated
assumptions must be analyzed first. The JPM is developed under the assumption of independence between the tidal
signal and SSs. Tawn and Vassie (1989) found that this assumption was false. Considering that this assumption may
be true under certain circumstances as proved by William et al. (2016) for the largest mid-latitude storm surges and
the corresponding tide. A tendency to overestimate sea levels, due to the fact that the correlation between tide SSs
has been ignored, was recognized in the literature (Pugh and Vassie, 1978, 1980; Walden et al., 1982). However, it
should be noticed that extreme levels such as the MSSs may be only very weakly dependent with high tides. This
constitutes a distinctive feature and advantage of the MSS based fitting procedure introduced in the present paper.
It is a major point of differentiation between the joint surge-tide probability procedures described in sections 2.
Furthermore, the hourly theoretical tides are in utmost cases considered as a realization of stationary process. This
assumption is the most critical one since sea levels are highly non-stationary due to the storm surge. As previously
argued to overcome this limitation, the variability arises from the SSs which can be considered as stationary over
the storms season for instance. For this argument to be less subjective, most high tides are similar in term of their
value and must be lower than the SS variation in extreme events.
The question one can ask is how to improve the modelling in such a way that the bias between the procedures using
SSSs and MSSs and the reference one is reduced as far as possible? Indeed, as depicted in figure 6, the second
procedure overestimates extreme sea levels for all the return periods (a maximizing envelope). The RLs estimates
for MSS based procedure are about 50 to 60 cm higher than those obtained when the SSS are used. The difference
between the upper and middle curves increase as the return period goes up. The difference is high for high return
periods. Inversely, the difference between the lower and middle curves increase as the return period goes down. The
difference is significant for lower return periods. It is noteworthy that the middle curve is supposed to represent the
RLs of reference. An objective answer to our question cannot in any case suggest a modification in the reference
method. Two methodological issues could provide us with solutions and answers to the question. First, the
dependence structure that exists between the high tide and the extreme SSs around the high tide could be modelled.
Extreme SSs one hour before the high tide, at the time of the high tide and one hour after can be used. A larger
window can likewise be used to consider the SSs around the high tide in a multivariate context.
A visual inspection with the scatter graphs and the Spearman's Rho numerical criteria have been used to measure
the statistical dependence between storm surges and tide at the moment of the high tide and around it (±1 hour). This
is useful when modeling the coincidence of the high tide with extreme storm surges, for instance. The Multivariate
frequency analysis consists in studying the dependence structure of two or more variables through a function that
depends on their marginal distribution functions. The multivariate theory is based on the mathematical concept of
copula (Sklar, 1959), which allows linking the distributions of the variables according to their degree of dependence.
More details can be found in (Salvadori and De Michele, 2004; Nelsen, 2006). A copula-based approach may be
used to consider this dependence. In the case of a copula of sea levels, no convolution is needed. The convolution
of SSs distribution with a density of tide permits to obtain a distribution of sea levels. This latter solution is proposed
herein as an alternative to the multivariate analysis using a copula.

The figure 7 shows the scatter graphs that provide a visual information about the dependence between the high tide
and the other variables (SSS, MSS and ESL). It can be concluded that the dependence with the two storm surge
variables SSS and MSS is weak and sufficiently low to consider the variables statistically independent. This finding
is supported by the Spearman's Rho coefficients and associated p-values are presented in Table 4. Indeed, to
determine if a correlation between the variables is significant or not, we need to conduct a Pearson correlation test
and compare the p-value to a significance level. In general, a significance level of 0.05 gives good results. This value
of $\alpha$ indicates that the risk of concluding that there is a correlation when in reality there is none is 5%. As a matter
of fact, the p-value is nothing other than the probability that the correlation coefficient is significantly different from
0. However, the Pearson coefficients are very close to zero for the SSS and MSS variables, and a zero coefficient
indicates that there is no linear dependence, whatever the p-value. A p-value is presented for each variable in Table
4. As Spearman's coefficients only correspond to one facet of dependence and to better analyse the association
between the SSs and high-tide, the Kendall's correlation coefficient is used, as well. It is often of interest in data
analysis and methodological research and similar to Spearman's correlation coefficient, it is designed to capture the
association between two variables. Results of the Kendall's tau test, also presented in Table 4, also support the
statistical significance of non-dependence between SSs and tide. The two sea level components (high tide and
extreme SSs) are then considered as independent random variables and the distribution of the total sea level can be
determined by convolution. Otherwise, a multivariate analysis based on the use of the copulas theory can be used.
**6. Further discussion**
As show in Figure 6, RLs obtained with the joint MSS-tide method are always higher than those using SSS. This is
consistent with the fact that the convolution process based on MSS uses only high water values for the tide density
(as it selects the maximum value of instantaneous SSs every 12 hours) and since MSS is always greater than or equal
to SSS. It is then logical to consider that the joint MSS-tide method is more conservative than the SSS based one.
As expected, figure 4 shows that ESL events at the right tail of the distribution, represented by the middle curve,
tend to be close to high SSS RLs which are dominated by the high tide. The results of this procedure confirm the
general finding highlighted in the literature (Fortunato et al., 2016; Haigh et al., 2016) that the return level
estimations obtained with the convolution tide-SSS are not adapted up to a certain return period (100 years in the
case of Le Havre). To overcome this problem, one can use the joint tide-MSS convolution method. Another solution
is to use an empirical method to define the left tail of the distribution and an extreme values analysis for the right
tail as stated by Tawn and Vassie (1989).
On the other hand, the current practices and statistical approaches to characterize the coastal flooding hazard by
estimating extreme storm surges and sea levels still have some weaknesses. Indeed, the combination of the tide and
the storm surge do not take into account several scenarios in particular those with a time-lag where the tide and the
storm surge could give likewise extreme sea levels. The choice of variables (high tide, SSSs, MSS, etc.) would be a
decisive step and an integral part of the logic behind the idea of combining the two phenomena. Interestingly, these
variables could also include other explanatory variables such as the time-lag between the two phenomena (tide and
SS). This time-lag would be an additional variable and it is defined as the difference of time of occurrence of the
second variable with respect to the first (e.g. time between a maximum storm surge and a high tide).

## 6.1 coincidence probability concept

Our interest to the probability of coincidence comes from our belief that a bias is introduced with the joint-MSS
convolution because it does not take into account the time difference between the maximum instantaneous SS and
the high tide. A probability of coincidence (i.e. the chance that a MSS occurs at the same time with high tide) can
be used to better characterize the extreme sea levels using the MSS. In the present paper, we are only interested in
the concept of the coincidence probability and the statistical dependence between MSS and tide at the moment of
the high tide and around it (±6 hours). An appropriate coincidence probability concept would then allow to better
estimate the probabilities and thus reduce the bias and bring the RLs closer to those obtained by the reference
method.
Let Δ be the time-lag between the high tide and the MSSs in each tide cycle. When considering coincidence, an
additional hazard curve, associated to the variable Δ can be built. The time-lag variable Δ, which would allow us to
compute a probability of coincidence, could be involved in a multivariate frequency analysis to consider the
dependence structure between the variables. It is also interesting to note that the probability of coincidence would
make it possible to conclude if the MSSs occur randomly in a tide cycle or not. The work must be performed for
many coastal systems with different physical properties to conclude whether or not there is a systematic temporal
dependence, and whether or not the extreme sea levels are overestimated if this is indeed the case.
As illustrated in the right panel of figure 2 the MSS can occur randomly somewhere around the high tide $M_n$. The
time difference between the MSS and the high tide is random as well. It is therefore quite legitimate to study it with
a frequency analysis method. Then a coincidence probability concept can be drawn as follows:

- Extract an independent sample of Δ
- Fit this sample with the appropriate distribution function. "Indeed, Δs is expressed in hours and it is not an extreme variable, it is bounded between -6H and 6H and can take any value with in this interval. There is then no tail of the distribution and the extreme value theory is not the appropriate framework to model this random variable. Thus, a uniform distribution would be a good fit for Δ.
- Use the desired probability to weight the probabilities of the MSSs, assuming that MSSs and Δ are independent. Many scenarios using many of these probabilities can be used in a probabilistic approach.

On the other hand and focusing on the statistical dependence, extreme SSs samples around the high tide (at the time
Δ of the high tide) was extracted. The largest window (±6 hour) centered on the time of the high tide was used and
the statistical dependence was then studied. Table 5 shows the Spearman's Rho measuring the statistical dependence
between storm surges and tide at the moment of the high tide and around it (±3 hour). It can be easily concluded that
the dependence between SSs and tides is very high around the time of high tide and it becomes weaker as delta
increases. As mentioned in the previous section, the dependence structure that exists between the MSSs around the
high tide could be modelled with copulas.

**6.2 The non-stationary context**

It is noteworthy that the climate change in the past and working in a non-stationary context can greatly affect and
invalidate the fit of the storm surge and sea level PDFs. Indeed, questions such as: what is the effect of potential
trends and jumps in the sea water level time series? And should this affect the results and its confidence? are fair
ones and justified. The non-stationary context is not covered by this paper because it moves us further away from
the main objective which is the use and the confrontation of different methods for quantifying the exceedance
probability of extreme sea levels. It could however be the object of another paper."

**7. Conclusions**

In the present paper, we provided a reasoning for the need, in a PFHA framework, to combine flood phenomena to
better characterize coastal flooding hazard. Few ideas have been proposed in the literature to tackle the combination
of tidal signals with extreme SSSs to estimate extreme sea levels. The present work supports these ideas, takes up
the tidal signals and SSSs convolution procedure and proposes a new procedure based on the MSSs useful to exploit
likewise the extreme SS events occurred during medium and low tide hours. Three fitting procedures have been
investigated. The first one employs the SSS as an explanatory variable with the tidal signals which are combined
with a JPM using a convolution of the tide density and the SSS distribution function. The second procedure uses the
same technique except that the MSSs are used instead of the SSSs. In the third approach, a frequency analysis is
performed using ESLs.
Another consideration in this paper was applying and illustrating these approaches on the example of the sea levels
in Le Havre, northwestern France, over the period 1971–2015. It may be noted that the methodology is not exemplary
developed for this case study; it applies to any site likely to experience a marine flooding. Fitting results in terms of
probability plots and extrapolated RLs using the three approaches are examined. Overall, the application has shown
that the RL estimates for MSS based convolution are quite different from those corresponding to the SSS based one.
Indeed, since MSS is always greater than or equal to SSS and since the convolution process using MSS selects the
maximum value of instantaneous SSs every tidal cycle, the RLs are systematically higher when the joint MSS-tide
method is used. But without properly tackling the probability of coincidence concept (i.e. the chance that a maximum
SS occurs at the same time with high tide) concept and the issue of temporal lag between tidal peaks and surge
peaks, the results will be probably always overestimated, which may not be useful for PFHA. the results of the MSS
based procedure are likely to contain a bias comparing to the direct statistics on ESLs which becomes more and
more important as return periods increase. In order to reduce this bias, the coincidence probability concept could be
helpful in making a more appropriate assessment of the risk using the MSS. On the other hand and if the MSS based
convolution is to be used, the application has shown the utility of modelling the dependence structure that exists
between the hourly SS values around the high tide (high tide ± 6 hours). Figure 6 shows that ESL events at the upper
tail of the distribution (the middle curve) tend to occur at the time of the high tide, as expected. The results of this

procedure confirm the general finding highlighted in the literature is that the RL estimations obtained with the convolution tide-SSS are not conclusive up to a certain return period (100 years in the case of Le Havre).

Perspective: An in-depth study could help to thoroughly improve the proposed procedure based on the use of MSS by developing the concept of coincidence and apply the developed concept on other sites of interest. A concept of coincidence and methodology to be developed should find additional applications for the assessment of risk associated to other combining flooding phenomena (e.g. pluvial flooding and storm surges).

**Author contributions:** Amine Ben Daoued wrote this paper with assistance from Yasser Hamdi, Nassima Mouhous-Voyneau and Philippe Sergent.

**Competing interests:** The authors declare that they have no conflict of interest.

**Acknowledgements** The authors would like to thank François Ropert, a research engineer (CEREMA, centre d'études et d'expertise sur les risques, l'environnement, la mobilité et l'aménagement) for his thoughtful comments and advices about copula theory application. The authors are grateful to the SHOM (Service Hydrographique et Océanographique de la Marine) for providing data.

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

**Table 1: Sea level and rainfall data sets**

| Type | Station | Period | Time step |
|------|---------|--------|-----------|
| Sea level | Harbour | 1971-2015 | 1h |


| | | | |
|------|---------|--------|-----------|
| Sea level | Harbour | 1971-2015 | 1h |


Table 2: POT thresholds for SSS, MSS and ESL variables

|  | SSS | MSS | ESL |
| --- | --- | --- | --- |
| Threshold u (m) | 0.59 | 0.75 | 0.81 |
| Poisson intensity $\lambda$ (average $N^{br}$ of events/year) | 1.45 | 1.13 | 2.83 |



**Table 3: Sea RLs and 95% confidence intervals for the three fitting procedures (in meters)**

| Method | T=10 | T=50 | T=100 | T=1000 |
|---|---|---|---|---|
| JPM-SSS | 8.31 (8.27-8.35) | 8.77 (8.72-8.82) | 8.89 (8.84-8.95) | 9.20 (9.07-9.32) |
| JPM-MSS | 8.84 (8.79-8.89) | 9.29 (9.22-9.36) | 9.42 (9.33-9.51) | 9.79 (9.58-10.01) |
| Frequency Analysis - ESL | 8.82 (8.74-8.91) | 8.99 (8.80-9.18) | 9.05 (8.79-9.31) | 9.22 (8.67-9.77) |



Table 4: Spearman's Rho coefficients (and associated p-values) as a measure of dependence between the tide and
the other variables

|  | SSSs ~ tide | MSSs ~ tide | ESL ~ tide |
|---|---|---|---|
| Spearman's test | -0.02<br>p-value = 0.0095 | -0.06<br>p-value < 2.2e-16 | 0.96<br>p-value < 2.2e-16 |
| Kendall's test | -0.01<br>p-value = 0.0074 | -0.05<br>p-value < 2.2e-16 | 0.83<br>p-value < 2.2e-16 |



Table 5: Spearman's Rho calculated between high tide and all the instantaneous surges in the tidal cycle

| Δ | -6 | -5 | -4 | -3 | -2 | -1 | +1 | +2 | +3 | +4 | +5 | +6 |
|---|----|----|----|----|----|----|----|----|----|----|----|----|
| High tide | 0.29 | 0.28 | 0.21 | 0.41 | 0.61 | 0.85 | 0.77 | 0.60 | 0.56 | 0.44 | 0.33 | 0.30 |


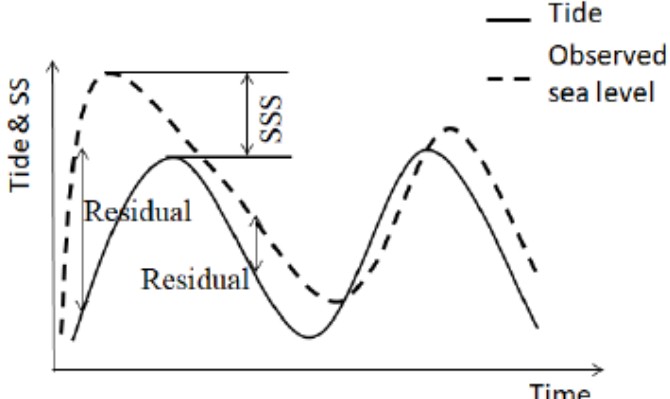

**Figure 1:** Definition and schematic representation of a skew storm surge


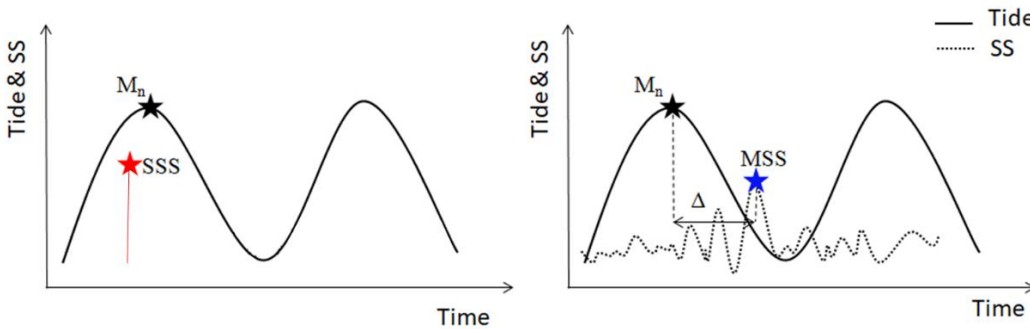

**Figure 2:** Illustration of tide and storm surge signals for the of joint surge-tide probability procedures: (left) skew
surge-tide combination; (right) maximum surge - tide combination


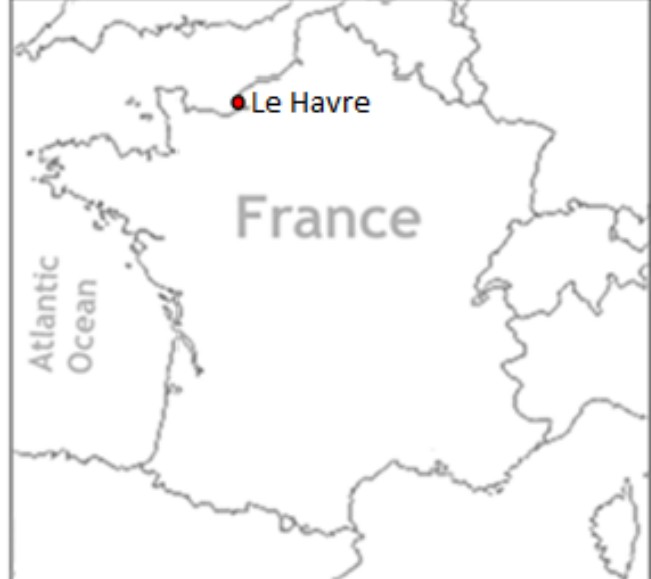

**Figure 3:** Case study (Le Havre): location map

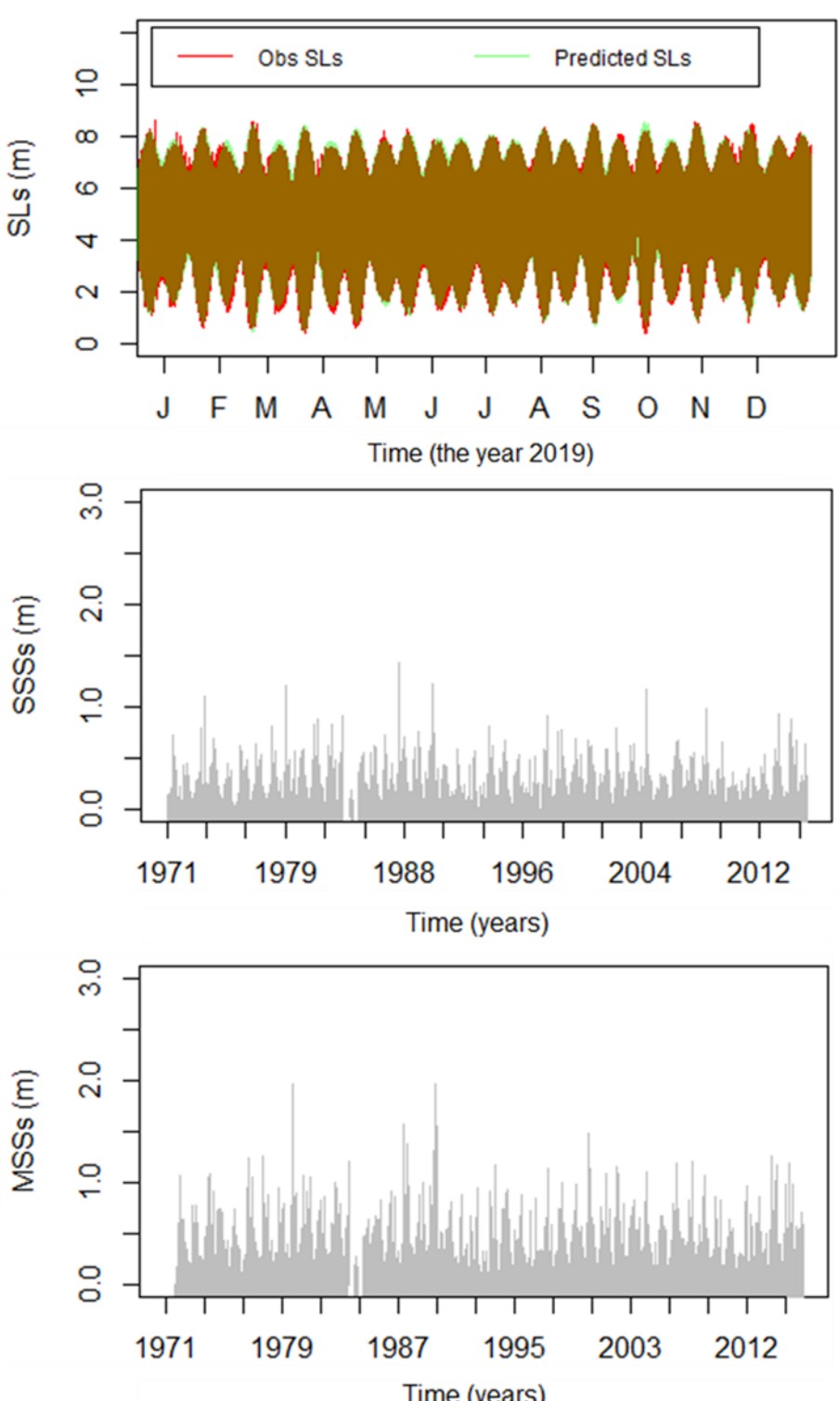

**Figure 4:** Studied time-series of Le Havre: (top) predicted and observed sea levels; (middle) SSSs data and
(bottom) the MSSs.

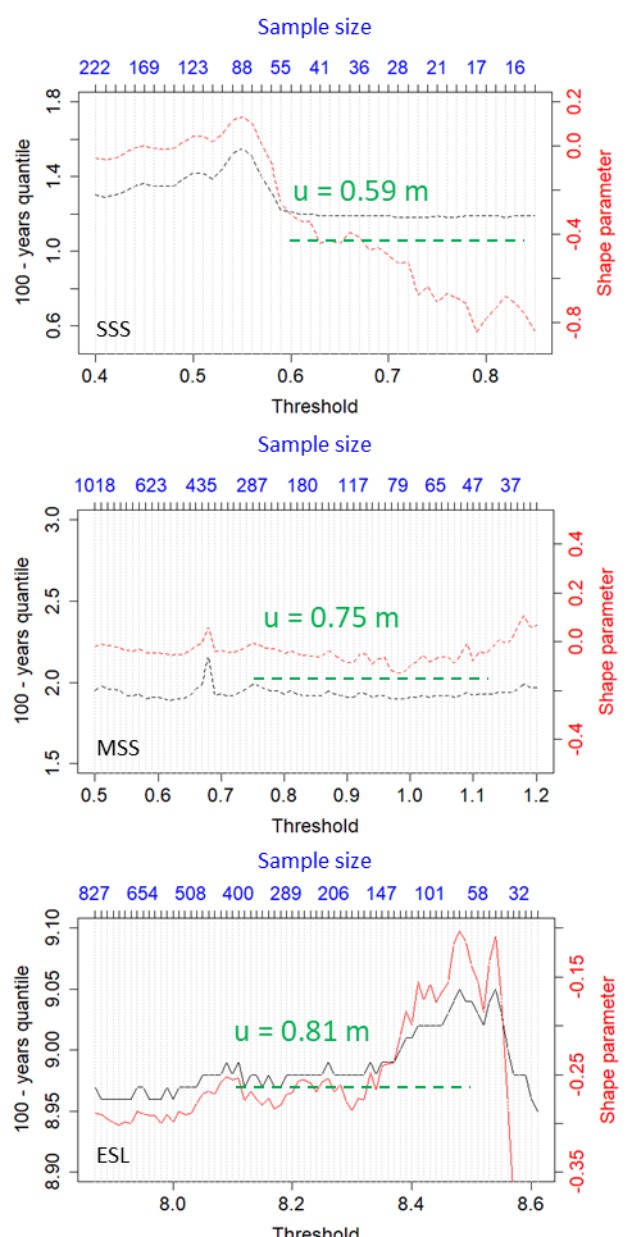


**Figure 5:** Stability plots for threshold selection: (top) SSSs, (middle) MSSs and (bottom) ESL


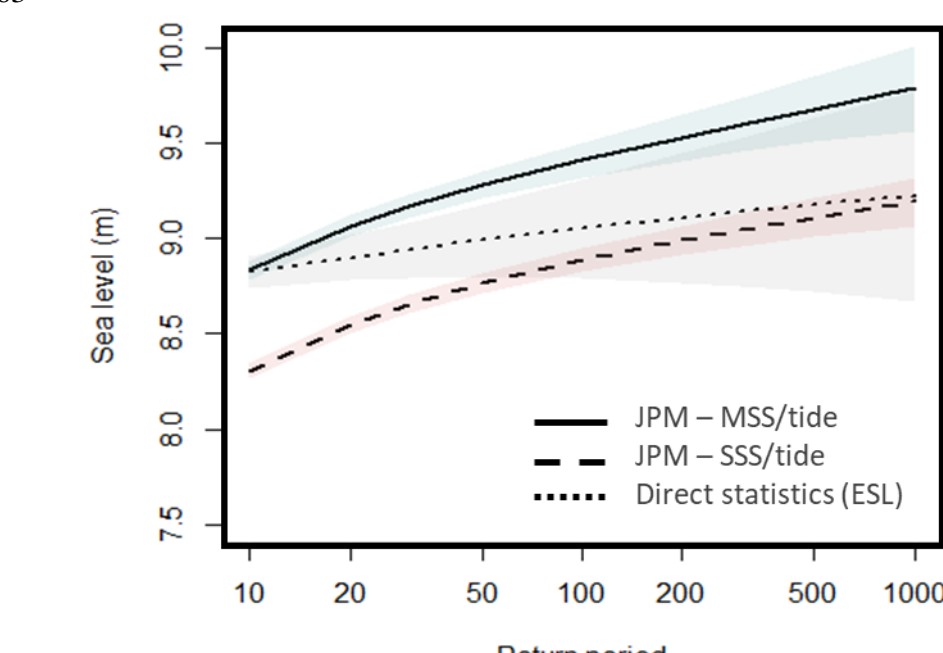


**Figure 6:** Sea level quantiles and confidence intervals

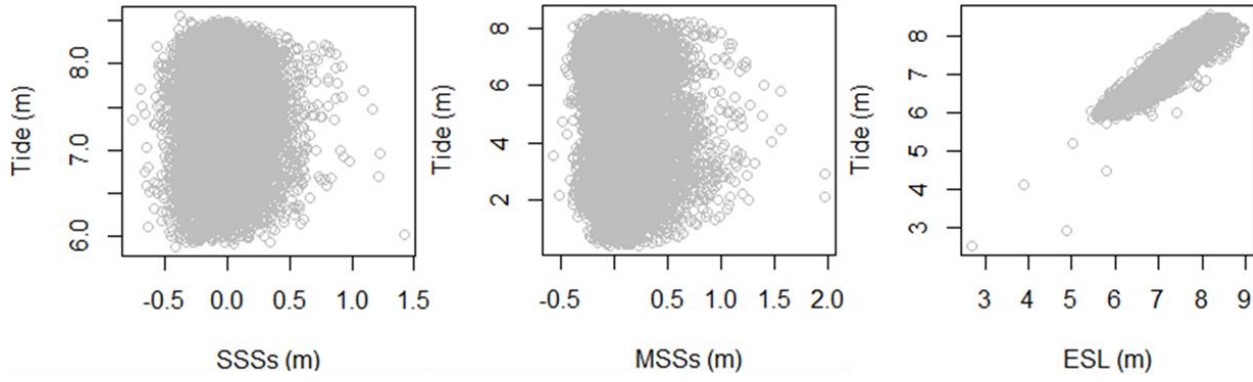

**Figure 7:** Analysis of the dependence between the tide and the SSSs, the MSSs and the ESL events