# Peer review of "Modelling dependence and coincidence of storm surges and high tide: Methodology, discussion and recommendations based on a simplified case study in Le Havre (France)"

_Natural Hazards and Earth System Sciences, 2019_

## Referee Comment (RC1) · Anonymous Referee #1 · 10 Apr 2020

The authors present a new method for quantifying the exceedance probability of extreme sealevels and compare it with two existing methods. One of these method is to directly sample extreme sealevels, while the other two methods indirectly construct the extreme sealevel distribution by applying a convolution between the astronomic high tide and either the skew surge or the maximum storm surge within a 12h window around high tide.

I fully agree with the authors that characterizing extreme sealevel distribution is of utmost importance to estimate coastal flood hazard and in this regard I find the idea conveyed by the title a relevant scientific and technical question. However, after read-

ing the manuscript, the reader is left with more questions and suggestions on how to study the dependence and coincidence of storm surges and the tide than answers. Therefore, I would not recommend publishing the manuscript in this current state unless some extensive extra work and analysis are performed. I also found the text hard to follow and I would suggest that the authors restructure their manuscript and carefully review the text for grammar and spelling mistakes.

General comments:

Too many details are sometimes given in points that are not further elaborated upon in the manuscript and on the opposite some critical information on the methods is missing. For example, the authors start the manuscript by discussing about nuclear power plant but this is not discussed further in the text other than they should not change the reference method. This seems to discredit the whole idea behind the need to compare and discuss different methods.

In the introduction, the authors discuss at length different types of other hazards happening in coastal areas (pluvial, fluvial floods) but this is not further looked into in the paper. If I understood correctly, the present study is on extreme sea levels and therefore extensively discussing about pluvial and fluvial floods seems out of the scope in my opinion. Similarly, it was not clear to me why the authors present in Table 1 the rainfall datasets if this is not used in this study.

There may be a general point to make that including statistical dependence is important to include when estimating (coastal) hazard but I am not sure why the authors put so much emphasis on this point if they don't themselves assess this statistical dependence in their selected case study. Throughout the paper, it is assumed that the tide and storm surge are independent but the authors never report on the validity of their assumption by reporting this statistical dependence. A good example of locations where this assumption might or might not be correct is given in Sterl, A., van den Brink, H., de Vries, H., Haarsma, R., and van Meijgaard, E.: An ensemble study of extreme

storm surge related water levels in the North Sea in a changing climate, Ocean Sci., 5, 369–378, https://doi.org/10.5194/os-5-369-2009, 2009.

At multiple points in the paper, the authors successively mention that dependence is not important but also that it could be important. These two statements, without further results or analysis, seem contradictory. For example page 3 – line 108-109: "Unlike to what is done very often in the literature, the question of dependency is not essential at all to combine phenomena in the present work. Indeed, as mentioned in the introductory section, tidal signals and SSs are independent." and later page 8 –line 283-284 "It has also been suggested that the questions of coincidence and dependency are essential for a combined tide and SS hazard analysis. "

The authors state that the maximum storm surge (MSS) can happen randomly somewhere within the tidal cycle. Again as showed in Sterl et al. (2009), I would argue that this is not the case and that the timing of the maximum storm surge is often closely related to physical properties of the coastal system. If this temporal dependence is present, I believe that the suggested method is likely to overestimate extreme sea levels

Table 2 and Figure 4 are not in line while I believe they should report the same values. When reading Table 2 for the 1000 year return period, one reads that MSS > ESL > SSS while when looking at Figure 4 the order is SSS > MSS > ESL. Based on my previous comment, I would suspect that the legend is Figure 4 was incorrectly labelled and that the highest curve shows the method based on the convolution with MSS.

In the discussion, the authors reflect on ways in which the possible dependence between the tide and storm surge and the timing between the latter could be included. The research presented here would greatly improve by actually doing these suggestions.

This paper would highly benefit from having more figures and analysis to make their point clear. For example, it would be interesting to see the studied time-series of Le

[Figure]

Havre, examples of extreme events, an analysis of the dependence between the tide and the skew surge and/or and/or the MSS and/or the ESL events.

The authors did not discuss nor report the effect of potential trends and jumps in the sea water level time series. They can greatly affect and invalidate the fit of the pdf and are often present in such time series.

Specific comments

The abstract would benefit from being more explicit: describe the three methods used and highlight some of the main differences (with numbers) and implications from these methods.

The extensive use of brackets makes the text at times hard to follow.

At the beginning of the results section, the authors present the R packages they used. In my opinion, this should belong to the Methods section.

Page 1 – line 11: "Tide and extreme SSs are considered as independent". Is this an assumption you made for this research or based on your results? If this is an assumption, then it seems contradictory to want to study the dependence but already assume that it is independent.

Page 1 – line 18: "It has also been suggested that the questions of coincidence and dependency are essential for a combined tide and SS hazard analysis." I would think that this is the question this paper is trying to answer.

Page 2 – line 53: "that the probability of failure (The probability of exceeding an extreme event)": Written in this way, it implies that the probability of failure is the equal to the exceedance probability and this is incorrect.

Page 2 – line 65: "SSS": At this point in the text, this acronym has not been defined yet.

Page 2 – line 71: "Salvadori and De Mechele". Please correct this typo for "Salvadori

and De Michele"

Page 3 – line 111:" On the other hand, it is commonly known today that the tidal signals can be predicted". Did the authors want to put the emphasis on the accuracy of the tidal predictions? Because the use of "today" implies that this is recent while this is actually known for some decades.

Page 4 – line 124: I think there is a mistake in equation 2 because fz(z) appears on both side of the equation. If I understood correctly, it should only be on the left-hand side of the equation

Page 4 – line 38-39: "Indeed, a SSS occurring with a high tide is more likely to induce a high sea level than an instantaneous SS occurring with any other tide." This statement is not clear to me. Can the authors elaborate to make their point?

Page 5 – line 150: "This feature makes the MSS a variable particularly useful for carrying out a PFHA exploring the entire tidal signal, not only the high tide ". If my understanding of the method is correct, each MSS value per tidal cycle is paired with the high tide value within this tidal cycle. If the MSS does not occur randomly within the tidal period, I believe this might highly overestimate your extreme sea levels which may not be useful for PFHA.

Page 5 – line 157: "As it can also be noticed for this reference procedure, the variable of interest would be the maximum sea level between 2 high-tide values. " Why do the authors mention "between 2 high-tide values"? Did you sample using a peaks over threshold method with some independence window criteria or using GEV?

Page 6 – line 187: please mention the final threshold selected, the resulting number of peaks used to fit the distribution in each case and add in supplementary the supplementary graphs.

Page 6 – line 193: "storm surge RLs": shouldn't this be water level return levels?

Page 6 – line 197: "with the delta method". Please briefly explain what is the delta

method and add appropriate references. I believe this is important since the authors go on to compare the width of the confidence interval.

Page 6 – line 218: " However, it should be noticed that extreme levels such as the MSSs may be only very weakly dependent." Can the authors elaborate on this sentence? I don't see why this would or would not be the case.

Page 7 – line 222: "This assumption is the most critical one since sea levels are highly non-stationary (due to the tide). " Shouldn't "tide" be replace with "storm surge" here?
* * *

---

## Referee Comment (RC2) · Anonymous Referee #2 · 13 Apr 2020

The present works investigates the dependence between tides and extreme surges and presents a new approach to quantify the exceedance probability of extreme sea levels; they compare it to two diferent methods, previously reported in other works. The idea is very interesting in particular in Le Havre site where the interaction between the astronomical components and the signal of surges is very important. The present work is relevant to study the extreme sea levels and assess the risks related to storms. However, many tasks have not been fully addressed and some explanations are required to improve this main bring of this work and make easier for the reader the understanding

of this approach. So, I recommend this work for publication after some moderate revision. Abstract. 'Tide and extreme SSs are considered as independent?' This sentence is disconnected from the previous one. What do you mean exactly? The previous study assumes the independence between SSs and tides? I don't understand the authors would study the dependence while they assume that "Tide and extreme SSs are considered as independent" in line 11 'Tide density? ' What do you mean by tide density The abstract does not reflect the main results of the work!! Introduction A very long sentence, difficult to understand! 'This goal is in line with the recent literature (e.g. Idier et al., 2012) challenging the use of the SSS and clearly demonstrates the importance of conducting extreme value analyses with maximum instantaneous ones. In order to achieve this goal, a third fitting procedure to estimate extreme sea levels using the maximum SS (MSS) between two consecutive 100 tides is introduced with an application so that it can be compared with the two first procedures.' It would be better if the choice of the Le Havre station can be justified: may be for the important interaction of the different driven forces induced by fluvial, tidal and wave activity. Methods What's MSS? What's JPM? It would be better if you can introduce clearly this!! Also, I have not understood how do you determine the SSs from the instantaneous measurements? The total sea level provided by tides is the sum of the SLR component, the long-term geological component, tides and the residual; Do you have considered the long-term components? Also, another important issue can be raised here. We can consider that the residual part as the surges, which is the dominant component sure but it's not the only one for this case Le Havre where the stochastic signal contains both surges and the fluvial effects! May be this should be signaled in the methods and the discussion. Again, I raise the necessity for readers, not expert if this area, to have the full description of the different abbreviations used!!! So, it will be better to introduce at the beginning of use each term! In relation with the use of the timeseries of LE Havre, how do you process the gaps? How do you have determined surges? By harmonic analyses? Line 150 oF page 5: "This feature makes the MSS a variable particularly useful for carrying out a PFHA exploring the entire tidal signal, not only the

high tide". The MSS value is paired with the high tide value within each tidal cycle? Then, the MSS could not occur always randomly within the tidal period. This approach could overestimate the extreme levels, I think line 157: As suggested, the variable of interest would be the maximum sea level between 2 high-tide values. So, my doubts is the following: Did you sample by the use of POT with the consideration of some independence window criteria or by the use of GEV?

Results Lines 253-251: variables are missing!

Page 6: what 's the final threshold selected and the peak number used to fit the distribution in each case Page 6 (line 193) the use of 'storm surge RLs' , do you refer to be water return levels? Page 6 (line 197) the delta method. Please can you explain what 's this?

The results section should be more detailed, may some illustrations are required in this stage!

The paper is very interesting and some improvements are required. I can recommend the publication of this paper after some revisions that I considered then as minor revision.

---

## Referee Comment (RC3) · Anonymous Referee #3 · 24 Apr 2020

The manuscript by Amine Ben Daoued and co-authors addresses an important issue for the modeling of exceedance probability of extreme sea-levels namely accounting for the dependence between storm surges and high tide. The authors present a new method that is compared to two existing ones (direct sampling of extreme sea-levels, and indirect construction of extreme sea-level distribution via a convolution between the astronomic high tide within a 12h window around high tide by using either the skew surge SSS or the maximum storm surge MSS). The methods are applied and compared on the Le Havre tide gauge.

Main comment.

[Figure]

The manuscript is well organized and the presentation is clear. Yet, several aspects should be clarified and further elaborated before publication (state of the art, details of the implementation, application to alternative real cases). Therefore, I recommend additional corrections by incorporating, if possible, the following recommendations.

Specific comments

1. State of the art.

I agree with the authors that most studies assume that "Tide and extreme SSs are considered as independent" (as stated in the abstract). Yet, this is not so systematic: I would reformulate by highlighting: "Most existing studies are generally based on the assumption that tide and extreme SSs are independent." Some studies (not cited by the authors) have addressed this problem with different approaches. These should be underlined in the introduction and further discussed by the authors.

In particular, - Coles, S., & Tawn, J. (2005). Seasonal effects of extreme surges. Stochastic Environmental Research and Risk Assessment, 19(6), 417-427; - Gouldby, B., Méndez, F. J., Guanche, Y., Rueda, A., & Mínguez, R. (2014). A methodology for deriving extreme nearshore sea conditions for structural design and flood risk analysis. Coastal Engineering, 88, 15-26. – see section 3.2; - Pirazzoli, P. A., & Tomasin, A. (2007). Estimation of return periods for extreme sea levels: a simplified empirical correction of the joint probabilities method with examples from the French Atlantic coast and three ports in the southwest of the UK. Ocean Dynamics, 57(2), 91-107;

Note that a more recent overview on the interaction with tides is provided by Idier et al. (2019): Idier, D., Bertin, X., Thompson, P., & Pickering, M. D. (2019). Interactions between mean sea level, tide, surge, waves and flooding: mechanisms and contributions to sea level variations at the coast. Surveys in Geophysics, 40(6), 1603-1630.

Finally, the beginning of the introduction is mainly focused on the problem of NPPs though the problem of tide-surge dependence is of interest in all applications of the

domain of coastal engineering. The authors should maybe either reformulate the introduction to be more general, or reflect the focus on NPPs directly in the title.

2. Details on the implementation.

The manuscript would benefit from additional implementation details (and figures) on the different steps of the proposed method. In particular, - Figure on the time-series of Le Havre with examples of MSS (SSS) and High tide sampling; - An empirical bivariate scatterplot High Tide versus MSS (or SSS); - Consider the possibility of statistical methods to estimate tide‐surge interaction like the analysis by Feng et al. (2015): Figure 6 or the chi‐square test described by Haigh et al., (2010); - Stability graphs for the choice of the threshold values; - Error estimates on the GPD parameters (line 206); - Further details on the delta method (page 6, line 197).

Besides, the authors refer to R packages: these references should be preferably located in the method section, together with additional formal details on the corresponding methods.

At the end of the discussion (page 7, from lines 340), the authors highlight some interesting alternative methods. These are very relevant and I must admit that after reading them, I wonder why the authors did not consider them in the frist place. Could the authors clarify this aspect?

References Feng, J., von Storch, H., Jiang, W., & Weisse, R. (2015). Assessing changes in extreme sea levels along the coast of C hina. Journal of Geophysical Research: Oceans, 120(12), 8039-8051. Haigh, I., N. Robert, and W. Neil (2010), Assessing changes in extreme sea levels: Application to the English Channel, 1900–2006, Cont. Shelf Res., 30, 1042–1055, doi:10.1016/j.csr.2010.02.002.

3. Application.

The application cases consists of one tide gauge, where the interaction between tide and surge is known to be high. Though the results on this site is useful to highlight

the possible pitfalls of neglecting the dependence as well as the differences between the three methods, the study would benefit from adding a new test case to discuss the influence of: - the strength of interactions for instance by choosing a site with less or no interaction (this location may done based on Idier et al., 2012 for instance); - the length of the time series.

4. Typo.

- Page 2 (line 65): "SSS" has not been introduced before.

- Page 2 (line 71): "Salvadori and De Mechele" should be "Salvadori and De Michele"

- Page 6 (line 193): "storm surge RLs": sea level RLs?

- Page 7 (line 255): the symbol after "this temporal difference" is not depicted properly in the manuscript pdf. The problem also appears in line 258 and 260.

―――――――――――――――――――――

---

## Referee Comment (RC4) · Anonymous Referee #4 · 27 Apr 2020

Review summary:

The main goal of the article, as claimed by its title, is to develop a methodology to model dependence and coincidence of storm surges and high tide, and to apply it to a case study in France. Although the objective is clear and laudable, the article proves to be disappointing as the initial goal is far from being reached.

The authors present and compare three methods to perform an Extreme Value Analysis (EVA) on sea levels on a single case study (Le Havre). One is based on a univariate direct approach (so called the reference method) as surge and tidal signals are not separated from each other but the total water level signal is instead considered as a

single random variable - let's call it method 1. The remaining two methods are based on an indirect approach considering that the probability density function (PDF) of total water level can be modeled as a convolution of tide and surge PDFs. The difference between the two lies in the use of a suitable random variable for "surge": one method uses the skew storm surge (SSS) - let's name it method 2, and the other one uses the maximum storm surge (MSS) - let's name it method 3. Authors claim that the use of MSS is in fact the novelty of the article.

Overall, I think the paper should be substantially improved before getting published. In any case, I recommend not publishing it as is. I think the novelty of the research is low and the conducted analysis is poor. Particularly, the article does not provide any method to model dependence and coincidence of storm surges and high tides, despite its title. At most it allows to highlight the issue of combining storm surge and tidal signals in an indirect approach of EVA.

I detail my review below, separating major from minor comments.

Major comments:

Introduction/State of the art:

- Although the article mentions some key references that investigated the issue of combining tides and SSs (e.g. Tawn and Vassie (1989), Dixon and Tawn (1994), Haigh et al (2010), Kergadallan et al (2014)), it is not clear how the present work differs from or compares with others, for example what is not addressed in those studies that will be in the present work. The authors also could have cited Mazas et al (2014) "Applying POT methods to the Revised Joint Probability Method for determining extreme sea levels", Coast. Eng. 91, 140-150. This study is in line with what is done in the present work. Mazas et al (2014) compared several methods to determine extreme sea levels on a single case study (Brest) using convolution of the tide and surge density functions, but testing hourly vs skew surges and two methods for handling tide-surge interaction. They also compared results with a direct approach, just as authors did. I think the

paper would benefit replacing the present work in this context and showing the novelty with respect to previous research.

- As the article focuses on extreme sea levels and indirect approach for EVA of sea levels, I think the entire introduction section should be revised to better document previous research in that domain (see for example the article of Batstone et al (2013)).

Methods:

- This section must be completed, as some basic information on EVA are not even mentioned. For instance, the authors do not describe the sampling method used in the analysis (either for SS or for total sea level marginals): do they use POT (as indicated in the results section line 187)? What extreme laws are used (Generalised Pareto Distribution or Generalised Extreme Value distribution?)? At least, the formula of the CDF should be provided, with appropriate definitions of parameters.

- I think that beginning of section 4 (results) from line 180 to line 195 should be included in the methods section.

- The method chosen by the authors for the indirect approach is a convolution of densities (tide and SS). But it is not clear to me if the tide density uses only high water values or the entire hourly time series. In addition, nothing is said about the derivation of tide density (which method is used? What is the duration of the sample used to derive the density?)

- Nothing is said either on the modelling of coincidence of storm surges and high tides in the methods section, although this is the title of the article.

Case study and data:

- Data characteristics (such as time step for the time series) should be given in the text (in addition to Table 1).

- The authors state that Le Havre is prone to marine and pluvial floods. In addition,

[Figure]

Table 1 relates characteristics of pluvial time series. Logically, I expected to see some compound events in the following with an appropriate method to tackle the issue. As pluvial data are not used in the present work, they should not be mentioned at all.

- There is a problem in the time span of tide gauge time series: 1971-2015 in the text VS 1938-2017 in Table 1.

Results:

- The authors write "the POT threshold selection process has been adapted to meet this criterion and the thresholds are, even though, checked regarding the stability graphs of the GPD parameters estimated with the maximum likelihood method." To appreciate the quality of the fit and to justify their choices, the authors should provide some plots.

- As mentioned above, I am not sure if the convolution process uses only high water values for the tide density. If this is the case (it should be according to Figure 2), and since MSS is always greater than or equal to SSS, it is logical that return levels (RLs) of method 3 are always higher than those obtained with method 2. Method 3 is actually conservative as it selects the maximum value of instantaneous SS every 12 hours (or so). But without properly tackling the issue of temporal lag between tidal peaks and surge peaks, the results are probably overestimated. The authors should discuss this point.

- There is a problem in the presentation of results: Table 2 and Figure 4 are not consistent. If I trust Table 2, then the reference curve (method 1) is the middle one. This is consistent with the text of the article (line 233). But still, I find the behavior of the RL curves in Figure 4 odd especially at lower return periods. For instance, according to previous research (see e.g. Kergadallan et al, 2014 or Mazas et al, 2014), method 2 should provide higher return levels than method 1.

- The results section would be improved with plots of return levels of SS (for both SSS and MSS).

Discussion:

- The authors write in line 244 "A copula-based approach may be used to study the dependence of instantaneous SSs (or sea levels)." What exactly does that mean? Is it a dependence in time (to model autocorrelation)? Copula would be used to model time dependence of SS? To take into account time dependence of SS or sea levels, extremal index could be considered (see e.g. Batstone et al, 2013).

- The paragraph in lines 248-252 is exactly what we expect to be presented in the article. The authors then propose a method to tackle the issue of coincidence but they do not try it. However, this should be the core of the article.

- I have some doubts about the proposed method. Although $\Delta$s is a random variable, it is not an extreme variable. Expressed in hours, it is bounded between 0 and 12 (or -6 and 6) and can take any value within this interval. There is no tail of the distribution and I do not think extreme value theory can apply in that case. Thus, speaking of return level of $\Delta$s does not make sense. In fact, I would say a uniform distribution would be a good fit for $\Delta$s.

- The statement in lines 260-261 is wrong. A frequency analysis does not imply an extreme value analysis.

Minor comments:

- L11: Authors write that "Tide and extreme SS are considered as independent." I think what authors mean is that in general, in most studies, tide and extreme SS are considered as independent. So this sentence should be modified as numerous studies have tried to tackle the issue of tide-surge dependence.

- L33: word to be deleted (in bold) : "The safety demonstration and protections **and** are. . ."

- L46-47: Probabilistic Flood Hazard Assessment. At least, the authors should mention the issue of multivariate return periods. Assessing flood hazard does not imply necessarily to compute the probabilities that one or more parameters are exceeded (see e.g. Salvadori et al (2011) "On the return period and design in a multivariate framework, Hydrol. Earth Sys. Sci., 15, 3293-3305).

- L51: "a river nuclear sites". Fragment unclear, consider revising.

- L53: spelling mistake (in bold) : "It is a common belief today that" . The probability of failure is not systematically the probability of exceeding an extreme event. This statement should be modified accordingly.

- L59 : "volume" does not seem appropriate for a river flood. I suggest to use the word "flow".

- L62: word is missing (in bold): "...marine flooding which is a combination of the tide (which can be predicted) with a SS." Defined like this, SS must also include the effect of waves (setup, runup). Since the effect of waves on total water level is not discussed nor mentioned in the article, this sentence needs rephrasing.

- L65: acronym SSS is not defined before.

- L71: Spelling mistake (in bold): "According to Salvadori and De Michele (2004)..."

- L80: Spelling mistake (in bold): Haigh et al (2010). Also the use of the word "recently" for a 10-year-old study is questionable.

- L87: I think a final point is missing after "distribution function of SSs".

- L91: reword (in bold): "GEV model is recommended"

- L92: the authors write "Based on the regional observations, the process of estimation of extreme water levels ..." Does that mean that this method (method 1) uses a regional frequency analysis ?

- L108: The authors write "Indeed, the SS is the main driver of coastal flood events". This is not true everywhere nor always. Coastal floods can occur from three main

mechanisms: overflowing, overtopping, breaching. Impacts of waves on structures are sometimes crucial and the main driver of coastal flooding. The statement must be reworded.

- L111: The authors state again (also in the introductory section) that "tidal signals and SSs are independent". This is not true, as shown in previous research (Idier et al, 2012; Batstone et al, 2013). The sentence must be reworded.

- L115-116 : the wording is awkward as extreme sea level is proposed as a variable to represent SS. This must be reworded.

- L124: Equation (2) is false: fZ(z) on the right hand side must be deleted.

- L126-127: I think there is a confusion here. The tide signal is clearly not a stationary stochastic process, but SS can be considered as so. As the authors write the opposite, they should clarify this point.

- L157-158: The sentence is not clear, I do not understand what is the variable of interest. Rewording should be considered.

- L174: Sentence is awkward and needs rephrasing.

- L193: Wording mistake (in bold): "and 1000-year sea level RLs".

- L205-206: it seems that GPD is used to describe the tails of the distributions of SS. This does not seem consistent with statement in L91 where GEV is recommended. The authors should clarify this point.

- L210-226: I find this paragraph unclear, I do not see what the authors want to say. I suggest making it clearer.

- L231-232: I think there is a wording mistake (in bold): "The difference is high for high return periods."

- L233: I think there is a wording mistake (in bold): "The difference is significant for

lower return periods"

- L236-239: I do not understand the end of the paragraph. The authors should clarify their statement.

- L257: POT is not an fitting method, it is a sampling method.

- L262: The authors write "figure 4 shows that extreme sea level events tend to occur at the time of the high tide". I do not see that in Figure 4. The authors should clarify their thought and better explain this result.

- L266-267: The end of section 5 is awkward and should be reworded. It seems that to overcome the problem of method 2, one just needs to follow Tawn and Vassie (1989). Then a question arises: why is method 3 necessary if method 2 limitation can be solved?

- L269-270: The first statement of the Conclusions section is a bit exaggerated. The authors should reword it.

- L277: I am not sure acronym ESL has been defined before.

- L281: spelling mistake (in bold): "**F**itting results in terms of probability..."

- L290: word missing (in bold)?: "...around the high tide (high tide **+/- 3 hours**).

- References should be listed alphabetically and homogenized.

- Figure 2: SSS is defined as the difference between maximum observed minus predicted sea levels. Therefore, it is a discretized time series and not a continuous one as pictured in Fig 2.

- Overall, English could be improved.

---

## Author Comment (AC1) · 8 Jul 2020

Dear Referee #1,

Thank you so much for reviewing our paper. The manuscript was modified to consider your constructive comments. In the following, a point-by-point response to your comments are presented.

Best regards,

The authors

[Figure]

Please also note the supplement to this comment:
https://www.nat-hazards-earth-syst-sci-discuss.net/nhess-2019-407/nhess-2019-407-AC1-supplement.pdf

————————————————————

[Figure]

**Supplement:**

Dear Referee #1,

Thank you so much for reviewing our paper.

The manuscript will be, therefore, modified to consider your constructive comments. In the following, a point-by-point response to your comments will be presented.

**Point-by-Point response / reviewer # 1**

Yasser Hamdi

| Comment | Responses to comments |
|---|---|
| General point 1: Too many details are sometimes given in points that are not further elaborated upon in the manuscript and on the opposite some critical information on the methods is missing. For example, the authors start the manuscript by discussing about nuclear power plant but this is not discussed further in the text other than they should not change the reference method. This seems to discredit the whole idea behind the need to compare and discuss different methods. | This is an interesting comment. The NPPs example is used as a motivation element. Yes, indeed, this work is done in a context of nuclear safety and review of the nuclear safety demonstration and protections. This was mentioned in the introduction section. It was also mentioned that the present work could be used to enrich safety verification approaches. It's also true that we don't aim to modify the reference method in the present work but attempt to propose other approaches, and simply confront all of them.

This is now clearly indicated in Sect. 1, page 2, lines 40-42

"The present work could be used to enrich safety verification methods by proposing other approaches and confronting them to the reference method currently used in the guide" |
| General comment 2: In the introduction, the authors discuss at length different types of other hazards happening in coastal areas (pluvial, fluvial floods) but this is not further looked into in the paper. If I understood correctly, the present study is on extreme sea levels and therefore extensively discussing about pluvial and fluvial floods seems out of the scope in my opinion. Similarly, it was not clear to me why the authors present in Table 1 the rainfall datasets if this is not used in this study. | We agree that discussing other flooding sources was a bit exaggerated. A part of this discussion is now removed.

Rainfall data characteristics are likewise removed from table 1. |
| There may be a general point to make that including statistical dependence is important to include when estimating (coastal) hazard but I am not sure why the authors put so much emphasis on this point if they don't themselves assess this statistical dependence in their selected case study. Throughout the paper, it is assumed that the tide and storm surge are independent but the authors never report on the validity of their assumption by reporting this statistical dependence. A good example of locations where this assumption might or might not be correct is given in Sterl, A., van den Brink, H., de Vries, | The comment is on matters of substance. Yes, indeed, it is always interesting to quantify the statistical dependence in a context of coastal flooding.

In another work, we combined the storm surge with other flood phenomena (riverine flooding and/or local rainfall, etc.) and the correlation of the variables of interest was evaluated. The statistical dependence was measured with a Chi-plot technique and non-parametric estimators (the upper tail dependence, for instance). This allowed us to decide modelling the dependence structure of the two variables using the copula theory (when they are dependent) and to only consider the univariate CDF's in case of independence. Indeed, we did not aim in the present work to show details on how evaluating the dependence in extreme value context.

Indeed, the general goal of the present paper is to characterize the hazard "coastal flooding" by combining the high-tide and extreme storm surges (SSSs & MSSs). A dependence analysis was conducted despite the fact that the study aims to use only the extreme values of these variables. Scatter graphs |

| | |
|---|---|
| H., Haarsma, R., and van Meijgaard, E.: An ensemble study of extreme storm surge related water levels in the North Sea in a changing climate, Ocean Sci., 5, 369–378, https://doi.org/10.5194/os-5-369-2009, 2009. | and the Spearman's Rho have been used to measure the statistical dependence between high-tide and extreme SSs. It was concluded that this dependence is weak and sufficiently low to consider the variables of interest dependents.

The following sentence is now used:

- In the Abstract (lines 11-12) : "Most existing studies are generally based on the assumption that high-tides and extreme SSs are independent."
- In the Methods section (lines 147-148): "Indeed, as mentioned in the introductory section and as it will be discussed later in this paper, extreme levels such as MSSs may be only very weakly dependent with high-tides."

The discussion section (lines 291-293 and 302-308 with figure 7) has been changed to add a discussion on the dependence analysis.

Another kind of dependence that caught our attention (but more important for the coincidence model) is the one between the high-tide and the other instantaneous storm surges around the high-tide (±6 hours). The Spearman's Rho was used as a measure of this statistical dependence (a further discussion section is now added to the paper). |
| At multiple points in the paper, the authors successively mention that dependence is not important but also that it could be important. These two statements, without further results or analysis, seem contradictory. For example page 3 – line 108-109: "Unlike to what is done very often in the literature, the question of dependency is not essential at all to combine phenomena in the present work. Indeed, as mentioned in the introductory section, tidal signals and SSs are independent." and later page 8 –line 283-284 "It has also been suggested that the questions of coincidence and dependency are essential for a combined tide and SS hazard analysis." | It was assumed in the present paper that the tide and storm surge are independent and a convolution model has been applied with a simple sum of them in the indirect method (with both, skew storm surges and instantaneous ones).

I must admit that there is a contradiction here. The two sentences are now modified:

Lines 145-147: "As it would be analyzed later in the discussion section, the dependency, in an extreme value context, is analyzed but not considered to combine the phenomena in the present work."

The second sentence has been removed to the beginning of the conclusion section.

"It has been suggested that the questions of combining tide and SSs is essential to better characterize the coastal flooding hazard."

In addition, as suggested by one of the reviewers, the sentence "Tide and extreme SSs are considered as independent" in the abstract is now replaced by: "Most existing studies are generally based on the assumption that high-tides and extreme SSs are independent." (lines 11-12). |
| | |
| The authors state that the maximum storm surge (MSS) can happen randomly somewhere within the tidal cycle. Again as showed in Sterl et al. (2009), I would argue that this is not the case and that the timing of the maximum storm surge is often closely related to physical properties of the coastal system. If this temporal dependence is present, I believe that the suggested method is likely to overestimate extreme sea levels. | Thank you for this comment and for suggesting the possible explanation. Yes it was assumed that a maximum storm surge can happen randomly somewhere within the tidal cycle. We didn't analyse the relationship that can exist between the timing of the MSS and the physical properties of the coastal system. We however recognize that considering this interaction between the timing of the MSSs and the coastal system is difficult to conduct and further investigations are here necessary. |
| Table 2 and Figure 4 are not in line while I believe they should report the same values. When reading Table 2 for the 1000 year return period, one reads that MSS > ESL > SSS while when looking at Figure 4 the order is SSS > MSS > ESL. Based on my previous comment, I would suspect that the legend is | Right, the legend is not correct. It is now correctly labelled. The table has the number 3 and the figure has the number 6 now. |

| | |
|---|---|
| Figure 4 was incorrectly labelled and that the highest curve shows the method based on the convolution with MSS. | |
| In the discussion, the authors reflect on ways in which the possible dependence between the tide and storm surge and the timing between the latter could be included. The research presented here would greatly improve by actually doing these suggestions. | Very interesting idea. We agree that this will greatly improve the present research. We propose adding a "further discussion" section to take up this reflection (the way in which the possible dependence between tide, storm surge and the timing between them). We included in this new section the following paragraphs (page ??, lines ??-??): |

[revised manuscript text omitted]

| | |
|---|---|
| This paper would highly benefit from having more figures and analysis to make their point clear. For example, it would be interesting to see the studied time-series of Le Havre, examples of extreme events, an analysis of the dependence between the tide and the skew surge and/or and/or the MSS and/or the ESL events. | More figures are now added:

- To the section case study: Figure 4. Studied time-series of Le Havre: (top) predicted and observed sea levels; (middle) SSSs data and (bottom) the MSSs.
- To the discussion section: Figure 7. Analysis of the dependence between the tide and the SSSs, the MSSs and the ESL events. |
| The authors did not discuss nor report the effect of potential trends and jumps in the sea water level time series. They can greatly affect and invalidate the fit | Yes, indeed working in a non-stationary context can greatly affect and invalidate the fit of the storm surge and sea level PDFs. We didn't consider it in this work because we think that it moves us further away from the main objective of the paper. It could however be the object of another paper. The |

| | |
|---|---|
| of the pdf and are often present in such time series. | following paragraph is now added to the further discussion section (lines: 363-369) |
| | "It is also noteworthy that the climate change in the past and working in a non-stationary context can greatly affect and invalidate the fit of the storm surge and sea level PDFs. Indeed, questions such as: what is the effect of potential trends and jumps in the sea water level time series? What would happen with projected sea level rise? Is the estimated return period affected? Should this affect the results and its confidence? are fair ones and perfectly justified. The non-stationary context is not covered by this paper because it moves us further away from the main objective which is the use and the confrontation of different methods for quantifying the exceedance probability of extreme sea levels. It could however be the object of another paper." |

**Minor comments**

| Comment | Response to reviewer |
|---|---|
| The abstract would benefit from being more explicit: describe the three methods used and highlight some of the main differences (with numbers) and implications from these methods. | Two sentences are now added to the abstract (lines 17-22 and 24-26) |
| The extensive use of brackets makes the text at times hard to follow. | Fixed |
| At the beginning of the results section, the authors present the R packages they used. In my opinion, this should belong to the Methods section. | the R packages we used are now presented in the Methods section |
| Page 1 – line 11: "Tide and extreme SSs are considered as independent". Is this an assumption you made for this research or based on your results? If this is an assumption, then it seems contradictory to want to study the dependence but already assume that it is independent. | It is rather an assumption for the Havre based on results. |
| Page 1 – line 18: "It has also been suggested that the questions of coincidence and dependency are essential for a combined tide and SS hazard analysis." I would think that this is the question this paper is trying to answer. | This sentence is now removed and replaced by the following one in the abstract just before talking about the case study:

 Lines 21-22: "The question we are trying to answer in this paper is then the coincidence and dependency essential for a combined tide and SS hazard analysis." |
| Page 2 – line 53: "that the probability of failure (The probability of exceeding an extreme event)": Written in this way, it implies that the probability of failure is the equal to the exceedance probability and this is incorrect. | "(The probability of exceeding an extreme event)" is now removed from the sentence. |
| Page 2 – line 65: "SSS": At this point in the text, this acronym has not been defined yet. | Fixed |
| Page 2 – line 71: "Salvadori and De Mechele". Please correct this typo for "Salvadori and De Michele" | OK |
| Page3–line111:" On the other hand, it is commonly known today that the tidal signals can be predicted". Did the authors want to put the emphasis on the accuracy of the tidal predictions? Because the use of "today" implies that this is recent while this is actually known for some decades. | The word "today" is now removed. |
| Page 4 – line 124: I think there is a mistake in equation 2 because fz(z) appears on both side of the equation. If I understood correctly, it should only be on the left-hand side of the equation | Right. The equation is now fixed. |

| | |
|---|---|
| Page4–line38-39: "Indeed, a SSS occurring with a high tide is more likely to induce a high sea level than an instantaneous SS occurring with any other tide." This statement is not clear to me. Can the authors elaborate to make their point? | This sentence is now simplified and replaced by the following one: Line 178: "Indeed, a SSS occurring with a high tide is likely to induce a high sea level" |
| Page 5 – line 150: "This feature makes the MSS a variable particularly useful for carrying out a PFHA exploring the entire tidal signal, not only the high tide ". If my understanding of the method is correct, each MSS value per tidal cycle is paired with the high tide value within this tidal cycle. If the MSS does not occur randomly within the tidal period, I believe this might highly overestimate your extreme sea levels which may not be useful for PFHA. | Yes indeed, if the MSS does not occur randomly within the tidal period. As mentioned earlier in our response to a general comments, the probability of coincidence would make it possible to conclude if the MSSs occur randomly in a tide cycle or not and it must be tested for many coastal systems (with different physical properties). |
| | On the other hand, overestimating extremes allow us to be more conservative in the nuclear safety field. But it is not our objective to overestimate the extreme sea levels. |
| | The following sentence (added to the conclusion section in response to a comment of another reviewer) takes up this view of point: |
| | Lines 385-390: "Indeed, since MSS is always greater than or equal to SSS and since the convolution process using MSS selects the maximum value of instantaneous SSs every tidal cycle, the RLs are systematically higher when the joint MSS-tide method is used. But without properly tackling the probability of coincidence concept (i.e. the chance that a maximum SS occurs at the same time with high tide) concept and the issue of temporal lag between tidal peaks and surge peaks, the results will be probably always overestimated, which may not be useful for PFHA." |
| Page 5 – line 157: "As it can also be noticed for this reference procedure, the variable of interest would be the maximum sea level between 2 high-tide values. " Why do the authors mention "between 2 high-tide values"? Did you sample using a peaks over threshold method with some independence window criteria or using GEV? | We extract the max sea level in each tidal cycle and then we use these data as raw data to extract extreme values with a classic POT frequency model. |
| Page 6 – line 187: please mention the final threshold selected, the resulting number of peaks used to fit the distribution in each case and add in supplementary the supplementary graphs. | The following sentence (with a table and a figure showing the POT frequency model characteristics) is now added at the end of the first paragraph of section results. |
| | Lines 238-240: "The POT model characteristics (threshold and associated average number of events per year) are presented in Table 2. The stability graphs for threshold selection are presented in Figure 5". |
| Page 6 – line 193: "storm surge RLs": shouldn't this be water level return levels? | Yes, it would be better. Changed. |
| Page 6 – line 197: "with the delta method". Please briefly explain what is the delta method and add appropriate references. I believe this is important since the authors go on to compare the width of the confidence interval. | The following sentence, with the appropriate reference, is now added to the end of the paragraph before the last one of the section results. |
| | Lines 251-253: "It is interesting to note that the delta method (*Ver Hoef, 2012*) is a classic technique in statistics for computing confidence intervals for functions of maximum-likelihood estimates. The variance of RL estimates are calculated using an asymptotic approximation to the normal distribution." |

| | |
|---|---|
| Page6–line218: "However, it should be noticed that extreme levels such as the MSSs may be only very weakly dependent." Can the authors elaborate on this sentence? I don't see why this would or would not be the case. | Because only one value per tidal cycle is extracted. |
| Page 7 – line 222: "This assumption is the most critical one since sea levels are highly non-stationary (due to the tide). " Shouldn't "tide" be replace with "storm surge" here? | Yes, indeed. Fixed. |

---

## Author Comment (AC2) · 8 Jul 2020

Dear Referee #2,

Thank you so much for reviewing our paper. The manuscript is modified to consider your constructive comments. In the following, a point-by-point response to your comments is presented.

Best regards,

The authors

[Figure]

Please also note the supplement to this comment:
https://www.nat-hazards-earth-syst-sci-discuss.net/nhess-2019-407/nhess-2019-407-AC2-supplement.pdf
* * *
Interactive
comment

[Figure]

**Supplement:**

Dear Referee #2,

Thank you so much for reviewing our paper.

The manuscript will be, therefore, modified to consider your constructive comments. In the following, a point-by-point response to your comments will be presented.

**Point-by-Point response / reviewer # 2**

Yasser Hamdi

| Comment | Responses to comments |
|---|---|
| Abstract: | |
| 'Tide and extreme SSs are considered as independent?' This sentence is disconnected from the previous one. What do you mean exactly? The previous study assumes the independence between SSs and tides? I don't understand the authors would study the dependence while they assume that "Tide and extreme SSs are considered as independent". | It was assumed in the present paper that the tide and storm surge are independent in an extreme value context and a convolution model has been applied with a simple sum of them in the indirect method (with both, skew storm surges and maximum instantaneous ones).

 Indeed, the general goal of the present paper is to characterize the hazard "coastal flooding" by combining the high-tide and extreme storm surges (SSSs & MSSs). A dependence analysis was conducted despite the fact that the study aims to use only the extreme values of these variables. Scatter graphs and the Spearman's Rho have been used to measure the statistical dependence between high-tide and extreme SSs. It was concluded that this dependence is weak and sufficiently low to consider the variables of interest dependents.

 The following sentence is now used:

 In the Abstract, as suggested by one of the reviewers, the sentence "Tide and extreme SSs are considered as independent" in the abstract is now replaced by: Lines 11-12: "Most existing studies are generally based on the assumption that high-tides and extreme SSs are independent."

 In addition, In the Methods section:

 Lines 147-148: "Indeed, as mentioned in the introductory section and as it will be discussed later in this paper, extreme levels such as MSSs may be only very weakly dependent with high-tides."

 The discussion section (lines 291-293 and 302-308 with figure 7) has been changed to add a discussion on the dependence analysis.

 Another kind of dependence that caught our attention (but more important for the coincidence model) is the one between the high-tide and the other instantaneous storm surges around the high-tide (±6 hours). The Spearman's Rho was used as a measure of this statistical dependence. A further discussion about this issue section is now added to the paper. |
| General comment 2: In the introduction, the authors discuss at length different types of other hazards happening in coastal areas (pluvial, fluvial floods) but this is not further looked into in the paper. If I understood correctly, the present study is on extreme sea levels and therefore extensively discussing about pluvial and fluvial floods seems | We agree that discussing other flooding sources was a bit exaggerated. A part of this discussion is now removed.

 Rainfall data characteristics are likewise removed from table 1. |

| | |
|---|---|
| out of the scope in my opinion. Similarly, it was not clear to me why the authors present in Table 1 the rainfall datasets if this is not used in this study. | |
| in line 11 'Tide density? ' What do you mean by tide density | The tide is not distributed randomly and its density can be used instead of a distribution function. |
| The abstract does not reflect the main results of the work!! | The main results of the work are now presented in the abstract (24-26) |

| Introduction | |
|---|---|
| A very long sentence, difficult to understand! 'This goal is in line with the recent literature (e.g. Idier et al., 2012) challenging the use of the SSS and clearly demonstrates the importance of conducting extreme value analyses with maximum instantaneous ones. In order to achieve this goal, a third fitting procedure to estimate extreme sea levels using the maximum SS (MSS) between two consecutive 100 tides is introduced with an application so that it can be compared with the two first procedures.' | I admit that the two sentences must be better expressed. Lines 126-128: "This goal is in line with the recent literature (e.g. Idier et al., 2012) challenging the use of the SSS and clearly demonstrates the importance of using the maximum instantaneous surges (MSSs) instead." and, Lines 128-130: "In order to achieve this goal, a third fitting procedure to estimate extreme sea levels using the MSSs between two consecutive tides is introduced with an application so that it can be compared with the two first procedures." |
| It would be better if the choice of the Le Havre station can be justified: may be for the important interaction of the different driven forces induced by fluvial, tidal and wave activity. | The following sentence is now added (the last of the introduction): Lines 139-140: "One of the most important features of this case study is the fact that the lower parts of Le Havre city are likely to be flooded by coastal floods and that the region has experienced important storms during the last few decades." |

| **Methods:** | |
|---|---|
| What's MSS? What's JPM? It would be better if you can introduce clearly this!! | Thank you for this comment. MSS is the maximum instantaneous storm surge between two high tides and JPM is the joint probability method (a convolution between tide density and the surge distribution function). These definitions are proposed in the introductory section. |
| Also, I have not understood how do you determine the SSs from the instantaneous measurements? The total sea level provided by tides is the sum of the SLR component, the long-term geological component, tides and the residual; Do you have considered the long-term components? | May be the reviewer means how do you determine the MSSs from the instantaneous measurements? AS defined in the introductory section, MSS is the maximum hourly storm surge in each tidal cycle. But if the reviewer means the skew storm surge (SSS), it is the difference between maximum observed level and maximum predicted level in each tidal cycle. It is defined in the introductory section as follows: Lines 89-90: "It is the difference between the highest observed level and the highest predicted one, for a same high tide. These maximum levels can occur at slightly different times." As it is a difference between two total levels, this definition takes only the water rise du the meteorological conditions. |
| Also, another important issue can be raised here. We can consider that the residual part as the surges, which is the dominant component sure but it's not the only one for this case Le Havre where the stochastic signal contains both surges and the fluvial effects! May be this should be signaled in the methods and the discussion. | The following sentence is now added to the method section. Lines 166-168: "It should also be noted that for the case Le Havre the residual part as the surges is not the only one and despite the fact that it is the dominant component, the stochastic signal also contains the fluvial effects." |
| Again, I raise the necessity for readers, not expert if this area, to have the full | A description of the different abbreviations used is now provided. |

| | |
|---|---|
| description of the different abbreviations used!!! So, it will be better to introduce at the beginning of use each term! | |
| In relation with the use of the time series of LE Havre, how do you process the gaps? | In the calculation of the effective duration, we take into account:

– The declustering tool used in independent events extract takes into account the presence of the gaps.
– The presence of gaps is also considered in the settings of the POT frequency model. Indeed, after threshold selection, the effective duration of observations (in years) is calculated by subtracting the gaps periods: the effective duration is then the ratio between number of days with observations and the average number of days in a year (365.25) |
| How do you have determined surges? By harmonic analyses? | The surges time series were already available. They were calculated in another framework. |
| Line 150 of page 5: "This feature makes the MSS a variable particularly useful for carrying out a PFHA exploring the entire tidal signal, not only the high tide". MSS value is paired with the high tide value within each tidal cycle? Then, the MSS could not occur always randomly within the tidal period. This approach could overestimate the extreme levels, I think. | Yes indeed, it could overestimate the extreme levels if the MSS does not occur randomly within the tidal period. The probability of coincidence would make it possible to conclude if the MSSs occur randomly in a tide cycle or not and it must be tested for many coastal systems (with different physical properties).

On the other hand, overestimating extremes, if it occurs, allows us to be more conservative in the nuclear safety field. But it is not our objective to overestimate the extreme sea levels. |
| line 157: As suggested, the variable of interest would be the maximum sea level between 2 high-tide values. So, my doubts is the following: Did you sample by the use of POT with the consideration of some independence window criteria or by the use of GEV? | The POT frequency model has been used after a declustering step. |
| Results Lines 253-251: variables are missing! | Ok. It's now fixed. |
| Page 6: what's the final threshold selected and the peak number used to fit the distribution in each case | These settings are now presented in table 2 (and figure 5). |
| Page 6 (line 193) the use of 'storm surge RLs' , do you refer to be water return levels? | Yes. Changed. |
| Page 6 (line 197) the delta method. Please can you explain what 's this? | The following sentence, with the appropriate reference, is now added to the end of the paragraph before the last one of the section results:

Lines 251-253: "It is interesting to note that the delta method (*Ver Hoef, 2012*) is a classic technique in statistics for computing confidence intervals for functions of maximum-likelihood estimates. The variance of RL estimates are calculated using an asymptotic approximation to the normal distribution." |
| The results section should be more detailed, may some illustrations are required in this stage! | More results and discussion are now presented in the paper. |

---

## Author Comment (AC3) · 8 Jul 2020

Dear Referee #3,

Thank you so much for reviewing our paper. The manuscript is modified to consider your constructive comments. In the following, a point-by-point response to your comments is presented.

Best regards,

The authors

[Figure]

Please also note the supplement to this comment:
https://www.nat-hazards-earth-syst-sci-discuss.net/nhess-2019-407/nhess-2019-407-AC3-supplement.pdf
* * *
[Figure]

**Supplement:**

Dear Referee #3,

Thank you so much for reviewing our paper.

The manuscript will be, therefore, modified to consider your constructive comments. In the following, a point-by-point response to your comments will be presented.

**Point-by-Point response / reviewer # 3**

Yasser Hamdi

**Specific comments**

| Comment 1- State of the art. | Our response |
|---|---|
| I agree with the authors that most studies assume that "Tide and extreme SSs are considered as independent" (as stated in the abstract). Yet, this is not so systematic: I would reformulate by highlighting: "Most existing studies are generally based on the assumption that tide and extreme SSs are independent." | Ok. The sentence is now changed. It is now replaced by the following one.

Lines 11-12: "Most existing studies are generally based on the assumption that high-tides and extreme SSs are independent." |
| Some studies (not cited by the authors) have addressed this problem with different approaches. These should be underlined in the introduction and further discussed by the authors.

In particular, - Coles, S., & Tawn, J. (2005). Seasonal effects of extreme surges. Stochastic Environmental Research and Risk Assessment, 19(6), 417-427; - Gouldby, B., Méndez, F. J., Guanche, Y., Rueda, A., & Mínguez, R. (2014). A methodology for deriving extreme nearshore sea conditions for structural design and flood risk analysis. Coastal Engineering, 88, 15-26. – see section 3.2; - Pirazzoli, P. A., & Tomasin, A. (2007). Estimation of return periods for extreme sea levels: a simplified empirical correction of the joint probabilities method with examples from the French Atlantic coast and three ports in the southwest of the UK. Ocean Dynamics, 57(2), 91-107; Note that a more recent overview on the interaction with tides is provided by Idier et al. (2019): Idier, D., Bertin, X., Thompson, P., & Pickering, M. D. (2019). Interactions between mean sea level, tide, surge, waves and flooding: mechanisms and contributions to sea level variations at the coast. Surveys in Geophysics, 40(6), 1603-1630. | We agree that adding more references would enrich the state of the art. This paragraph is now added to introductory section:

Lines 64-81: "The problem of the surge-tide interactions has been addressed in the literature for many regions and with different approaches (Coles and Tawn, 2005; Gouldby et al., 2014; Pirazzoli, 2007; Idier et al., 2012; Idier et al., 2019). It was shown that tide–surge interactions can be relevant in several regions. The tide–surge interactions at the Bay of Bengal (corresponding to the effect of the tide on atmospheric surge and vice versa) were analyzed by Johns et al., (1985) and Krien et al., (2017). They showed that tide–surge interactions in shallow areas of this large deltaic zone are in the range ±0.6m occurred at a maximum of 1 to 2 hours after low tide. Similar results were obtained by Johns et al. (1985), Antony and Unnikrishnan (2013) and more recently Hussain and Tajima (2017). Focusing on the English channel, Idier et al. (2012) used shallow water model to make surge computations with and without tide for two selected events (November 2007 North Sea and March 2008 Atlantic storms). The authors concluded that the instantaneous tide–surge interaction are significant in the eastern half of the English Channel, reaching values of 74 cm in the Dover Strait, which is about half of maximal storm surges induced by the same events. They also concluded that Skew surges are tide-dependent, with negligible values (less than 5 cm) over a large portion of the English Channel, but reaching several tens of centimeters in some locations such as the Isle of Wight and Dover Strait. More recently, Idier et al. (2019) have investigated the interactions between the sea level components (sea level rise, tides, storm surges, etc.) and the tide effect on atmospheric storm surges is among the main interactions investigated in their review. The authors stated that the studies, and other ones, converge to highlight that tide–surge interactions can produce tens of centimeters of water level at the coast." |

| | The following references are now added to the references list: |
|---|---|
| | • Antony, C. and Unnikrishnan A.S.: Observed characteristics of tide–surge interaction along the east coast of India and the head of Bay of Bengal. Estuar. Coast. Shelf. Sci. 131, 6–11. doi: 10.1016/j.ecss.2013.08.004, 2013.
• Coles, S., Tawn, J.: Seasonal effects of extreme surges. Stoch Environ Res Ris Assess, 19, 417–427, doi: 10.1007/s00477-005-0008-3, 2005.
• Gouldby, B., Mendez, F., Guanche, Y., Rueda, A. and Mínguez, R.: A methodology for deriving extreme nearshore sea conditions for structural design and flood risk analysis. Coastal Engineering. 88, 15–26. doi: 10.1016/j.coastaleng.2014.01.012, 2014.
• Hussain M.A. and Tajima Y.: Numerical investigation of surge–tide interactions in the Bay of Bengal along the Bangladesh coast. Nat Hazards 86(2):669–694. Doi: 10.1007/s11069-016-2711-4, 2017.
• Krien Y, Testut L, Islam AKMS, Bertin X, Durand F, Mayet C, Tazkia AR, Becker M, Calmant S, Papa F, Ballu V, Shum CK, Khan ZH Towards improved storm surge models in the northern Bay of Bengal. Cont. Shelf Res. 135, 58–73, doi: 10.1016/j.csr.2017.01.014, 2017.
• Pirazzoli, P.A. and Tomasin, A.: Estimation of return periods for extreme sea levels: a simplified empirical correction of the joint probabilities method with examples from the French Atlantic coast and three ports in the southwest of the UK. Ocean Dynamics, 57(2), 91-107, 2007.
• Idier D, Dumas F, Muller H Tide–surge interaction in the English channel. Nat Hazard Earth Sys, 12, 3709–3718, doi : 10.5194/nhess -12-3709-2012, 2012.
• Idier, D., Bertin, X., Thompson, P. and Pickering, M.D.: Interactions Between Mean Sea Level, Tide, Surge, Waves and Flooding: Mechanisms and Contributions to Sea Level Variations at the Coast. Surv Geophys 40, 1603–1630, doi: 10.1007/s10712-019-09549-5, 2019. |
| Finally, the beginning of the introduction is mainly focused on the problem of NPPs though the problem of tide-surge dependence is of interest in all applications of the domain of coastal engineering. The authors should maybe either reformulate the introduction to be more general, or reflect the focus on NPPs directly in the title. | The first paragraph of the introduction is now modified to be more general and consider other coastal facilities. |
| **2. Details on the implementation.** | |
| The manuscript would benefit from additional implementation details (and figures) on the different steps of the proposed method. In particular, - Figure on the time-series of Le Havre with examples of MSS (SSS) and High tide sampling; - An empirical bivariate scatterplot High Tide versus MSS (or SSS); - Consider the possibility of statistical methods to estimate tideˇ A˘Rsurge interaction like the analysis by Feng et al. (2015): Figure 6 or the chiˇ A˘Rsquare test described by Haigh et al., (2010); - Stability graphs for the choice of the threshold values; - Error estimates on the GPD parameters | Many figures (time series, scatter plots, stability plots for threshold selection, etc.) are now added.

- To the case study section: Figure 4. Studied time-series of Le Havre: (top) predicted and observed sea levels; (middle) SSSs data and (bottom) the MSSs.
- To the discussion section: Figure 7. Analysis of the dependence between the tide and the SSSs, the MSSs and the ESL events.
- To the results section: Figure 5. Stability plots for threshold selection: (top) SSSs, (middle) MSSs and (bottom) ESL. |

| | |
|---|---|
| (line 206); - Further details on the delta method (page 6, line 197). | |
| Besides, the authors refer to R packages: these references should be preferably located in the method section, together with additional formal details on the corresponding methods. | Ok. Done. |
| At the end of the discussion (page 7, from lines 340), the authors highlight some interesting alternative methods. These are very relevant and I must admit that after reading them, I wonder why the authors did not consider them in the frist place. Could the authors clarify this aspect? | It is now clarified in a further discussion section. |
| **3. Application.** | |
| The application cases consists of one tide gauge, where the interaction between tide and surge is known to be high. Though the results on this site is useful to highlight | It is a good idea. This is a thesis paper and only Le Havre case study was used in this thesis. |

**Minor comments:**

| Comment | Response to reviewer |
|---|---|
| - Page 2 (line 65): "SSS" has not been introduced before. | Ok. Fixed. |
| - Page 2 (line 71): "Salvadori and De Mechele" should be "Salvadori and De Michele" | Ok. Fixed. |
| - Page 6 (line 193): "storm surge RLs": sea level RLs? - | OK. Done. |
| Page 7 (line 255): the symbol after "this temporal difference" is not depicted properly in the manuscript pdf. The problem also appears in line 258 and 260. | OK. Fixed. |

---

## Author Comment (AC4) · 8 Jul 2020

Dear Referee #4,

Thank you so much for reviewing our paper. The manuscript is modified to consider your constructive comments. In the following, a point-by-point response to your comments is presented.

Best regards,

The authors

[Figure]

Please also note the supplement to this comment:
https://www.nat-hazards-earth-syst-sci-discuss.net/nhess-2019-407/nhess-2019-407-AC4-supplement.pdf
* * *
[Figure]

**Supplement:**

Dear Referee #4,

Thank you so much for reviewing our paper.

The manuscript will be, therefore, modified to consider your constructive comments. In the following, a point-by-point response to your comments will be presented.

**Point-by-Point response / reviewer # 4**

Yasser Hamdi

**Major comments:**

| Comment 1- Introduction/State of the art | Our response |
|---|---|
| Although the article mentions some key references that investigated the issue of combining tides and SSs (e.g. Tawn and Vassie (1989), Dixon and Tawn (1994), Haigh et al (2010), Kergadallan et al (2014)), it is not clear how the present work differs from or compares with others, for example what is not addressed in those studies that will be in the present work.

The authors also could have cited Mazas et al(2014)"Applying POT methods to the Revised Joint Probability Method for determining extreme sea levels", Coast. Eng. 91, 140-150. This study is in line with what is done in the present work. Mazas et al (2014) compared several methods to determine extreme sea levels on a single case study (Brest) using convolution of the tide and surge density functions, but testing hourly vs skew surges and two methods for handling tide-surge interaction. They also compared results with a direct approach, just as authors did. I think the paper would benefit replacing the present work in this context and showing the novelty with respect to previous research. | This is an interesting comment.

• The work of Mazas et al. (2014) is now cited in the introduction section with a brief comparison to the present work.
• More details and references about the tide-surge dependence are now added to the introduction section.
• More details about the work performed by Kergadallan et al., (2014) and how it differs from the present work is now added to the introduction section.
• The fact that Idier et al. (2012) and Kergadallan et al., (2014) performed the work with skew surges (and not the MSSs) is a main point of difference with the present work. The following sentence was already in the introduction:
Lines 126-128: "This goal is in line with the recent literature (e.g. Idier et al., 2012, Kergadallan et al., 2014) challenging the use of the SSS and clearly demonstrates the importance of using the maximum instantaneous surge (MSS) instead."

We agree that adding more references would enrich the state of the art. These two paragraphs are now added to introductory section:

1. Lines 130-135: "Mazas et al., (2014) proposed a review of tide-surge interaction methods and applied a POT frequency model (with the GPD and Poisson distribution functions) to the family of JPM-type approaches for determining extreme sea level values in a single case study (Brest). The authors focused on the use of a mixture model for the surge component, which allows probabilities to be quantified for the entire range of sea level values, not just for the extreme ones, which is not the case here in the present paper."

2. Lines 64-81: "The problem of the surge-tide interactions has been addressed in the literature for many regions and with different approaches (Coles and Tawn, 2005; Gouldby et al., 2014; Pirazzoli, 2007; Idier et al., 2012; Idier et al., 2019). It was shown that tide–surge interactions can be relevant in several regions. The tide–surge interactions at the Bay of Bengal (corresponding to the effect of the tide on atmospheric surge and vice versa) were analyzed by Johns et al., (1985) and Krien et al., (2017). They showed that tide–surge interactions in shallow areas of this large deltaic zone are in the range ±0.6m occurred at a maximum of 1 |

| | |
|---|---|
| | to 2 hours after low tide. Similar results were obtained by Johns et al. (1985), Antony and Unnikrishnan (2013) and more recently Hussain and Tajima (2017). Focusing on the English channel, Idier et al. (2012) used shallow water model to make surge computations with and without tide for two selected events (November 2007 North Sea and March 2008 Atlantic storms). The authors concluded that the instantaneous tide–surge interaction are significant in the eastern half of the English Channel, reaching values of 74 cm in the Dover Strait, which is about half of maximal storm surges induced by the same events. They also concluded that Skew surges are tide-dependent, with negligible values (less than 5 cm) over a large portion of the English Channel, but reaching several tens of centimeters in some locations such as the Isle of Wight and Dover Strait. More recently, Idier et al. (2019) have investigated the interactions between the sea level components (sea level rise, tides, storm surges, etc.) and the tide effect on atmospheric storm surges is among the main interactions investigated in their review. The authors stated that the studies, and other ones, converge to highlight that tide–surge interactions can produce tens of centimeters of water level at the coast." |
| As the article focuses on extreme sea levels and indirect approach for EVA of sea levels,I think the entire introduction section should be revised to better document previous research in that domain (see for example the article of Batstone et al (2013)). | As mentioned in the previous point, the introduction section has been revised and research in the combined tide-surge field and EVA are better documented. The following references are now used in the introduction section and added to the references list.

 • Antony, C. and Unnikrishnan A.S.: Observed characteristics of tide–surge interaction along the east coast of India and the head of Bay of Bengal. Estuar. Coast. Shelf. Sci. 131, 6–11. doi: 10.1016/j.ecss.2013.08.004, 2013.
 • Coles, S., Tawn, J.: Seasonal effects of extreme surges. Stoch Environ Res Ris Assess, 19, 417–427, doi: 10.1007/s00477-005-0008-3, 2005.
 • Gouldby, B., Mendez, F., Guanche, Y., Rueda, A. and Mínguez, R.: A methodology for deriving extreme nearshore sea conditions for structural design and flood risk analysis. Coastal Engineering. 88, 15–26. doi: 10.1016/j.coastaleng.2014.01.012, 2014.
 • Hussain M.A. and Tajima Y.: Numerical investigation of surge–tide interactions in the Bay of Bengal along the Bangladesh coast. Nat Hazards 86(2):669–694. Doi: 10.1007/s11069-016-2711-4, 2017.
 • Krien Y, Testut L, Islam AKMS, Bertin X, Durand F, Mayet C, Tazkia AR, Becker M, Calmant S, Papa F, Ballu V, Shum CK, Khan ZH Towards improved storm surge models in the northern Bay of Bengal. Cont. Shelf Res. 135, 58–73, doi: 10.1016/j.csr.2017.01.014, 2017.
 • Pirazzoli, P.A. and Tomasin, A.: Estimation of return periods for extreme sea levels: a simplified empirical correction of the joint probabilities method with examples from the French Atlantic coast and three ports in the southwest of the UK. Ocean Dynamics, 57(2), 91-107, 2007.
 • Idier D, Dumas F, Muller H Tide–surge interaction in the English channel. Nat Hazard Earth Sys, 12, 3709–3718, doi : 10.5194/nhess -12-3709-2012, 2012.
 • Idier, D., Bertin, X., Thompson, P. and Pickering, M.D.: Interactions Between Mean Sea Level, Tide, Surge, Waves and Flooding: Mechanisms and Contributions to Sea Level Variations at the Coast. Surv Geophys 40, 1603–1630, doi: 10.1007/s10712-019-09549-5, 2019. |

| | • Mazas, F., Kergadallan, X., Garat, P. and Hamm L.: Applying POT methods to the Revised Joint Probability Method for determining extreme sea levels. Coastal Engineering 91 140–150, 2014. |
|---|---|
| **2. Methods:** | |
| This section must be completed, as some basic information on EVA are not even mentioned. For instance, the authors do not describe the sampling method used in the analysis (either for SS or for total sea level marginals): do they use POT (as indicated in the results section line 187)? What extreme laws are used (Generalised Pareto Distribution or Generalised Extreme Value distribution?)? At least, the formula of the CDF should be provided, with appropriate definitions of parameters. | A sampling method sub-section is now added to the methods section (lines 198-206 ):

 **"2.4 The sampling method**

 The Peaks-Over-Threshold (POT) sampling method is used conduct the frequency analyses in the present work. Commonly considered as an alternative to the annual maxima method, the POT method models the peaks exceeding a relatively high threshold. The distribution of these peaks converge to the Generalized Pareto Distribution (GPD) theoretical distribution. In addition, the threshold leads to a sample more representative of extreme events. However, the threshold selection is subjective and an optimal threshold is difficult to obtain. Indeed, a too low threshold can introduce a bias in the estimation because some observations may not be extreme data and this violates the principle of the extreme value theory. On another hand, the use of a too high threshold reduces the sample size. "

 In addition, the section results contains now figures and more details about the frequency model settings (lines 236-240 with Table 2 and Figure 5). |
| I think that beginning of section 4 (results) from line 180 to line 195 should be included in the methods section. | Ok. It is now in the methods section. |
| The method chosen by the authors for the indirect approach is a convolution of densities (tide and SS). But it is not clear to me if the tide density uses only high water values or the entire hourly time series. In addition, nothing is said about the derivation of tide density (which method is used? What is the duration of the sample used to derive the density?) | All the tide density is used in the model but only the high tide is summed to SSSs and MSSs in order to calculate extreme sea levels.

 On the other hand, we used predicted tides already available for the Havre harbour, with the same duration of the sea level data set. Studied time-series of Le Havre (observed and predicted tide, SSSs and MSSs) are now better presented in the case study section (with plots). |
| Nothing is said either on the modelling of coincidence of storm surges and high tides in the methods section, although this is the title of the article. | A further discussion section take up all these aspects is now added to the paper (lines 309-369). |
| **3. Case study and data:** | |
| Data characteristics (such as time step for the time series) should be given in the text (in addition to Table 1). | As mentioned in table 1, the time step is one hour. The word "hourly" is now added in the case study section (line 227):

 "The 1971-2015 observed and predicted hourly sea levels … " |
| The authors state that Le Havre is prone to marine and pluvial floods. In addition, Table 1 relates characteristics of pluvial time series. Logically, I expected to see some compound events in the following with an appropriate method to tackle the issue. As pluvial data are not used in the present work, they should not be mentioned at all. | It was a mistake. Pluvial data is now removed from the table 1. |
| There is a problem in the time span of tide gauge time series: 1971-2015 in the text VS 1938-2017 in Table 1. | You are right. The time span of tide gauge time series is now fixed. |

| Results: | |
|---|---|
| The authors write "the POT threshold selection process has been adapted to meet this criterion and the thresholds are, even though, checked regarding the stability graphs of the GPD parameters estimated with the maximum likelihood method." To appreciate the quality of the fit and to justify their choices, the authors should provide some plots. | Stability plots for threshold selection are now presented in the results section and discussed (lines 236-240 with Table 2 and Figure 5). |
| As mentioned above, I am not sure if the convolution process uses only high water values for the tide density. If this is the case (it should be according to Figure 2), and since MSS is always greater than or equal to SSS, it is logical that return levels (RLs) of method3 are always higher than those obtained with method 2. Method3 is actually conservative as it selects the maximum value of instantaneous SS every 12 hours (or so). But without properly tackling the issue of temporal lag between tidal peaks and surge peaks, the results are probably overestimated. The authors should discuss this point. | Yes indeed, the approach using the MSS variable could overestimate the extreme levels if the MSSs does not occur randomly within the tidal period. The probability of coincidence (considering time lag between tidal and surge peaks) would make it possible to conclude if the MSSs occur randomly in a tide cycle or not and it must be tested for many coastal systems (with different physical properties). |
| | On the other hand, overestimating extremes, if it occurs, allows us to be more conservative in the nuclear safety field. But it is not our objective to overestimate the extreme sea levels. |
| | The following paragraph is now added to the discussion section (first paragraph): |
| | Lines 310-314: "As shown in Figure 6, RLs obtained with the joint MSS-method are always higher than those using SSS. This is consistent with fact that the convolution process based on MSS uses only high water va for the tide density (as it selects the maximum value of instantaneous every 12 hours) and since MSS is always greater than or equal to SSS. then logical to consider that the joint MSS-tide method is m conservative than the SSS based one.." |
| | And in the conclusion as well: |
| | Lines 385-389: "Indeed, since MSS is always greater than or equal to SSS and since the convolution process using MSS selects the maximum value of instantaneous SSs every tidal cycle, the RLs are systematically higher when the joint MSS-tide method is used. But without properly tackling the probability of coincidence concept (i.e. the chance that a maximum SS occurs at the same time with high tide) concept and the issue of temporal lag between tidal peaks and surge peaks, the results will be probably always overestimated." |
| There is a problem in the presentation of results: Table 2 and Figure 4 are not consistent. If I trust Table 2, then the reference curve (method 1) is the middle one. This is consistent with the text of the article (line 233). But still, I find the behavior of the RL curves in Figure 4 odd especially at lower return periods. For instance, according to previous research (see e.g. Kergadallan et al, 2014 or Mazas et al, 2014), method 2 should provide higher return levels than method 1. The results section would be improved with plots of return levels of SS (for both SSS and MSS). | Yes indeed, there is a mistake in the legend. It is now fixed. |
| **Discussion:** | |
| The authors write in line 244 "A copula-based approach may be used to study the dependence of instantaneous SSs (or sea levels)." What exactly does that mean? Is it a dependence in time (to model autocorrelation)? Copula would be used to model time dependence of SS? To | Here, we are rather talking about dependence between variables. |
| | The sentence is now changed to: |
| | Lines 297-298: "A copula-based approach may be used to consider this dependence." |

| | |
|---|---|
| take into account time dependence of SS or sea levels, extremal index could be considered (see e.g. Batstone et al, 2013). | |
| The paragraph in lines 248-252 is exactly what we expect to be presented in the article. The authors then propose a method to tackle the issue of coincidence but they do not try it. However, this should be the core of the article. | A further discussion section presenting the coincidence between SSs and high tide is now added to the paper. |
| I have some doubts about the proposed method. Although $\Delta s$ is a random variable, it is not an extreme variable. Expressed in hours, it is bounded between 0 and 12 (or -6 and6) and can take any value with in this interval. There is no tail of the distribution and I do not think extreme value theory can apply in that case. Thus, speaking of return level of $\Delta s$ does not make sense. In fact, I would say a uniform distribution would be a good fit for $\Delta s$. | Very good issue! Yes indeed, non-extreme distributions could be more appropriate for the lag time variable. The following sentence is now added in the further discussion section. Lines 250-253: "Indeed, $\Delta s$ is expressed in hours and it is not an extreme variable, it is bounded between -6 and 6H and can take any value with in this interval. There is then no tail of the distribution and the extreme value theory is not the appropriate framework to model this random variable. Thus, a uniform distribution would be a good fit for $\Delta s$." The RLs term is removed and the sentence is now changed to: Lines 354-355: "Use the desired probability to weight the probabilities of the MSSs, assuming that MSSs and $\Delta s$ are independent. Many scenarios using many of $\Delta s$ probabilities can be used in a probabilistic framework." |
| The statement in lines 260-261 is wrong. A frequency analysis does not imply an extreme value analysis. | Ok. |

**Minor comments:**

| Comment 1- Introduction/State of the art | Our response |
|---|---|
| L11: Authors write that "Tide and extreme SS are considered as independent." I think what authors mean is that in general, in most studies, tide and extreme SS are considered as independent. So this sentence should be modified as numerous studies have tried to tackle the issue of tide-surge dependence. | Ok. Done. |
| L33: word to be deleted (in bold): "The safety demonstration and protections and are…" | Ok. The word "are" is now deleted. |
| L46-47: Probabilistic Flood Hazard Assessment. At least, the authors should mention the issue of multivariate return periods. Assessing flood hazard does not imply necessarily to compute the probabilities that one or more parameters are exceeded (see e.g. Salvadori et al (2011) "On the return period and design in a multivariate framework, Hydrol. Earth Sys. Sci., 15, 3293-3305). | Thank you for this comment. It is interesting. The following sentence is now added (but at a later paragraph in the introduction section). Lines 91-94: "As more than one explanatory variable are often used in a PFHA and in case these variables are dependent, the dependency structure must be modeled and a consistent theoretical framework must be introduced for the calculation of the return periods and design quantiles with multivariate analysis based on Copulas (e.g. Salvadori et al., 2011). Indeed,…" Also, the following reference is now added to the references list: "Salvadori, G., De Michele, C., and Durante, F.: On the return period and design in a multivariate framework, Hydrol. Earth Syst. Sci., 15, 3293–3305, https://doi.org/10.5194/hess-15-3293-2011, 2011." |
| L51: "a river nuclear sites". Fragment unclear, consider revising. | Ok. Replaced by: "… flood hazard for nuclear sites located alongside rivers…" (line 58). |

| | |
|---|---|
| L53: spelling mistake (in bold) : "It is a common belief today that" . | Ok. Corrected. |
| The probability of failure is not systematically the probability of exceeding an extreme event. This statement should be modified accordingly. | Changed. |
| L59 : "volume" does not seem appropriate for a river flood. I suggest to use the word "flow". | The sentence is already deleted as suggested by another reviewer. |
| L62: word is missing (in bold): "...marine flooding which is a combination of the tide (which can be predicted) with a SS." | Ok. Corrected. |
| Defined like this, SS must also include the effect of waves (setup, runup). Since the effect of waves on total water level is not discussed nor mentioned in the article, this sentence needs rephrasing. | The following sentence is now added two sentences later:

 Line 86: "It should be noted that the effect of waves (runup and setup) on total water level is not discussed in the present paper." |
| L65: acronym SSS is not defined before. | Ok. It is now defined. |
| L71: Spelling mistake (in bold): "According to Salvadori and De Michele (2004)..." | Ok. Fixed. |
| L80: Spelling mistake (inbold): Haigh et al (2010). Also the use of the word "recently" for a 10-year-old study is questionable. | Yes, sorry about that. Fixed. |
| L87: I think a final point is missing after "distribution function of SSs". | Right! A final point is now added. |
| L91: reword (in bold): "GEV model is recommended" | Ok. Corrected. |
| L92: the authors write "Based on the regional observations, the process of estimation of extreme water levels..." Does that mean that this method (method1) uses a regional frequency analysis ? | No, here we talk about the FEMA study which recommend working in a regional scale... with regional frequency analysis.

 Otherwise, in reply to the question: all the methods use at-site observations. |
| L108: The authors write "Indeed, the SS is the main driver of coastal flood events". This is not true everywhere nor always. Coastal floods can occur from three main mechanisms: overflowing, overtopping, breaching. Impacts of waves on structures are sometimes crucial and the main driver of coastal flooding. The statement must be reworded. - | We then suggest: "Indeed, the SS is one of the main driver of coastal flood events". (line 84) |
| L111: The authors state again (also in the introductory section) that "tidal signals and SSs are independent". This is not true, as shown in previous research (Idier et al, 2012; Batstone et al, 2013). The sentence must be reworded. | Replaced with:

 Lines 147-148: "Indeed, as mentioned in the introductory section and as it will be discussed later in this paper, extreme levels such as MSSs may be only very weakly dependent with high-tides." |
| L115-116 : the wording is awkward as extreme sea level is proposed as a variable to represent SS. This must be reworded. | The sentence is now changed to:

 Lines (152-153): "So the question that arises here is which variable of interest can be used to better characterize coastal flooging?" |
| L124: Equation (2) is false: fZ(z) on the right hand side must be deleted. | Ok. Corrected. |
| L126-127: I think there is a confusion here. The tide signal is clearly not a stationary stochastic process, but SS can be considered as so. As the authors write the opposite, they should clarify this point. | You are right, there is a confusion here. The sentence is now changed to:

 Lines 164-166: "The hourly SS is often considered as a stationary stochastic process, since meteorological and seasonal effects give rise to series of SSs randomly |

| | |
|---|---|
| | distributed in time, but this is not the case of the hourly theoretical tide signals." |
| L157-158: The sentence is not clear, I do not understand what is the variable of interest. Rewording should be considered. | The sentence is now changed to: Lines (196-197): "The maximum sea level between 2 high-tide values is the variable of interest used for this reference procedure." |
| L174: Sentence is awkward and needs rephrasing. | Sentence changed to: Lines (229-230): "One of the most important features of Le Havre is the fact that it is subject to marine submersions and instabilities" |
| L193: Wording mistake (in bold): "and 1000-year sea level RLs". | OK. Corrected. |
| L205-206: it seems that GPD is used to describe the tails of the distributions of SS. This does not seem consistent with statement in L91 where GEV is recommended. The authors should clarify this point. | The GEV was recommended by FEMA (2004)… but the GPD is used herein. It is now clarified in the Introduction section. (Line 120) |
| L210-226: I find this paragraph unclear, I do not see what the authors want to say. I suggest making it clearer. | The paragraph is now modified and we hope that it is clearer now. |
| L231-232: I think there is a wording mistake (in bold): "The difference is high for high return periods." | You are right. Corrected. |
| L233: I think there is a wording mistake (in bold): "The difference is significant for lower return periods" | Ok. Corrected. |
| L236-239: I do not understand the end of the paragraph. The authors should clarify their statement. | Ok. The statement is now modified and we hope it is clearer now. |
| L257: POT is not an fitting method, it is a sampling method. | Ok. But The sentence is already changed. |
| L262: The authors write "figure 4 shows that extreme sea level events tend to occur at the time of the high tide". I do not see that in Figure 4. The authors should clarify their thought and better explain this result. | The sentence is: "Furthermore, figure 4 shows that extreme sea level events (the right tail of the distribution: the middle curve) tend to occur at the time of the high tide, as expected." The paragraph is now removed to a further discussion section and the sentence is now replaced by: Lines 314-315: "As expected, figure 4 shows that ESL events at the right tail of the distribution, represented by the middle curve, tend to be close to high SSS RLs which are dominated by the high-tide." |
| L266-267: The end of section 5 is awkward and should be reworded. It seems that to overcome the problem of method 2, one just needs to follow Tawn and Vassie (1989). Then a question arises: why is method 3 necessary if method 2 limitation can be solved? | This is a good comment. The sentence is now changed to: Lines 318-320: "To overcome this problem, one can use the joint tide-MSS convolution method. Another solution is to use an empirical method to define the left tail of the distribution and an extreme values analysis for the right tail as stated by Tawn and Vassie (1989)." |
| L269-270: The first statement of the Conclusions section is a bit exaggerated. The authors should reword it. | The first sentence is now replaced by the following: Lines 371-372: "In the present paper, we provided a reasoning for the need, in a PFHA framework, to combine flood phenomena to better characterize coastal flooding hazard." |

| | |
|---|---|
| L277: I am not sure acronym ESL has been defined before. | It was defined in the introduction section. I also define it in the abstract. |
| L281: spelling mistake (in bold): "Fitting results in terms of probability..." | Ok. |
| L290: word missing (in bold)?: "...around the high tide (high tide +/- 3 hours). | Ok. |
| References should be listed alphabetically and homogenized. | Ok. |
| Figure 2: SSS is defined as the difference between maximum observed minus predicted sea levels. Therefore, it is a discretized time series and not a continuous one as pictured in Fig 2. | The figure 2 is now changed. |
| Overall, English could be improved. | I hope English is now better. |

---

## Author Response (AR2)

Dear Referee,

Thank you so much for reviewing our paper for a second time.

The manuscript will be, therefore, modified to consider your comments. In the following, a point-by-point response to your comments will be presented.

**Point-by-Point response**

Yasser Hamdi

The corrections made by the authors improved the presentation of the results (by specifying the details of the implementation for instance), and enriched the discussion. I thank the authors for taking into account my recommendations. Yet, there are some residual moderate corrections that should be tackled before publication:

- The authors clearly state their research question in the abstract, lines 22-23 "The question we are trying to answer in this paper is then the coincidence and dependency essential for a combined tide and SS hazard analysis", but they only give partial answers to it by highlighting the differences of the MSS-based with the SSS-based approach. Some elements of the conclusions could be used in the abstract to clarify this aspect. As far as I understand the conclusions, the authors outline the difference between a procedure using MSS and another one using SSS, and recommend a copula-based dependence modelling to overcome the MSS-induced bias. If correct, this could be better highlighted in the abstract.

Authors' response: Yes indeed, a bias is introduced in the MSS based procedure comparing to the direct statistics on extreme sea levels and this bias is more important for high return periods. The assessment of the risk using the MSS will be more appropriate if a coincidence probability concept is used. The other solution is the use of copulas to consider the dependence structure (SSs ~ high tide ± 6 hours). We agree with the reviewer that it is an interesting point and must be also presented in the abstract. The following two sentences are now added to the abstract:
" The results brought to light a bias in the MSS based procedure comparing to the direct statistics on SLs and this bias is more important for high return periods. It was also concluded that an appropriate coincidence probability concept, considering the dependence structure between SSs, is needed for a better assessment of the risk using the MSS. "

- The title suggests that the authors propose a modelling approach of the coincidence, but the conclusion clearly state that (lines 455-456): "But without properly tackling the probability of coincidence concept …", and the authors propose some modelling recommendations (for instance based on copula) – Sect. 6. The title should better reflect this aspect, for instance: "Modelling dependence and coincidence of storm surges and high-tide: Methodology, Discussion and Recommendations based on a simplified case study in Le Havre (France)."

Authors' response: Right, the probability of coincidence is presented in the paper as a recommendation. The following title is now used:
"Modelling dependence and coincidence of storm surges and high-tide: Methodology, Discussion and Recommendations based on a simplified case study in Le Havre (France)."

- The authors highlight the absence of correlation between high tide and SSS (or MSS) by analyzing the Spearman's correlation coefficients (Table 2). They draw their conclusion on the low order of magnitude (line 343): I recommend deriving the p-value to support the statistical non-significance. Besides, I recommend analyzing also the Kendall's coefficients to have another proxy of dependence (Spearman's coefficients only correspond to one facet of dependence).

Authors' response: Yes indeed, to determine if a correlation between the variables is significant or not, we need to conduct a Pearson correlation test and compare the p-value to a significance level. In general, a significance level of 0.05 gives good results. This value of $\alpha$ indicates that the risk of concluding that there is a correlation when in reality there is none is 5%. As a matter of fact, the p-value is nothing other than the probability that the correlation coefficient is significantly different from 0. However, the Pearson coefficients are very close to zero for the SSS and MSS variables, and a zero coefficient indicates that there is no linear relationship, whatever the p-value. A p-value is presented for each variable in Table 4. As Spearman's coefficients only correspond to one facet of dependence and to better analyse the association between the SSs and high-tide, the Kendall's correlation coefficient is used, as well. It is often of interest in data analysis and methodological research and similar to Spearman's correlation coefficient, it is designed to capture the association between two variables. Results of the Kandall's tau test, also presented in Table 4, also support the statistical significance of non-dependence between SSs and tide.

This paragraph is now added to the section 5 (in the last paragraph). Table 4 was also updated with p-values and Kendall's tau results.

- line 343: 'independents' should be 'independent'
Authors' response: OK. corrected

- High et al. (2016) should le be Haigh et al. (2016).
Authors' response: OK. corrected

- Figure 4 – top : what is the time series indicated by brown lines ?
Authors' response: it is the superposition of the two colors green and red.

[revised manuscript text omitted]

|---|---|---|---|
| tide
Spearman's test | -0.02
p-value = 0.0095 | -0.06
p-value < 2.2e-16 | 0.96
p-value < 2.2e-16 |
| Kendall's test | -0.01
p-value = 0.0074 | -0.0463
p-value < 2.2e-16 | 0.8327754
p-value < 2.2e-16 |

Table 5: Spearman's Rho calculated between high tide and all the instantaneous surges in the tidal cycle

| Δ | -6 | -5 | -4 | -3 | -2 | -1 | +1 | +2 | +3 | +4 | +5 | +6 |
|---|----|----|----|----|----|----|----|----|----|----|----|----|
| High tide | 0.29 | 0.28 | 0.21 | 0.41 | 0.61 | 0.85 | 0.77 | 0.60 | 0.56 | 0.44 | 0.33 | 0.30 |

[Figure]

**Figure 1:** Definition and schematic representation of a skew storm surge

[Figure]

**Figure 2:** Illustration of tide and storm surge signals for the of joint surge-tide probability procedures: (left) skew
surge-tide combination; (right) maximum surge - tide combination

[Figure]

**Figure 3:** Case study (Le Havre): location map

[Figure]

**Figure 4:** Studied time-series of Le Havre: (top) predicted and observed sea levels; (middle) SSSs data and
(bottom) the MSSs.

[Figure]

**Figure 5:** Stability plots for threshold selection: (top) SSSs, (middle) MSSs and (bottom) ESL

[Figure]

**Figure 6:** Sea level quantiles and confidence intervals

[Figure]

**Figure 7:** Analysis of the dependence between the tide and the SSSs, the MSSs and the ESL events